# Spin-exciton coupling modified by interfacial magnetic interactions in a van der Waals heterostructure

Weican Lan[1,6], Chaocheng Liu [1,6] ✉, Yajuan Feng [1,6], Ruiqi Liu[1], Yafei Chu[1], Lu Cheng[1], Chao Wang [1], Huijuan Wang[2], Minghui Fan[3], Zixun Zhang[3], Yuran Niu [4], Jheng-Cyuan Lin[5], Francesco Maccherozzi [5], Hengli Duan [5] ✉ & Wensheng Yan [1] ✉

Excitons are primary elementary excitations in solids that present both fundamental interest and technological importance, showing great potential for photospintronic and quantum transduction applications. The emerging coherent collective excitations in two-dimensional antiferromagnetic semiconductors raise prospects for spin-exciton interactions and multifield control schemes. However, realizing the arbitrary manipulation of excitonic quantum states, while preserving the inherent dynamic and response advantages of antiferromagnetic nature remains challenging. Here we achieve bidirectional modulation of the CrSBr exciton energy via interfacial interaction-modified spin-exciton coupling in a $CrSBr/Fe_3GaTe_2$ heterostructure. Compared with pristine CrSBr, the photoluminescence peaks in the heterostructure can exhibit blueshift and redshift corresponding to 6.1% and 8.6% of the total bandwidth, respectively. We reveal that the interfacial charge-transfer-driven magnetic coupling in the heterostructure effectively enhances the magnetic anisotropy and the exchange interaction of CrSBr, thereby stabilizing its antiferromagnetic spin configuration, suppressing interlayer electron-hole recombination, and ultimately leading to an anomalous blueshift of the exciton emission. Our findings demonstrate an approach for bidirectionally modulating exciton energy in two-dimensional antiferromagnetic semiconductors, which provides substantial flexibility in device design and offers an avenue for potential wavelength control in quantum information and optoelectronic technologies.

Excitons, as quantized bound states of electron-hole pairs in semiconductors, have become a central research focus for next-generation optoelectronic devices and quantum technologies[1–5] due to their unique photoexcitation and energy transfer properties[6,7]. Exciton states can strongly couple with external fields at the level of spin, valley, and orbital degrees of freedom, translating their quantum characteristics into macroscopically controllable functionalities[8–13]. Consequently, the incorporation of magnetism and semiconductor

[1]National Synchrotron Radiation Laboratory, University of Science and Technology of China, Hefei, China. [2]Experimental Center of Engineering and Material Science, University of Science and Technology of China, Hefei, China. [3]Hefei National Research Center for Physical Sciences at the Microscale, University of Science and Technology of China, Hefei, China. [4]MAX IV Laboratory, Lund University, Lund, Sweden. [5]Diamond Light Source, Harwell Science and Innovation Campus, Didcot, United Kingdom. [6]These authors contributed equally: Weican Lan, Chaocheng Liu, Yajuan Feng. ✉e-mail: chaochengliu@ustc.edu.cn; hengli.duan@diamond.ac.uk; ywsh2000@ustc.edu.cn

electronics[14–16] offers exciting prospects for exploring fundamental excitation phenomena, as well as driving the unification of charge and spin manipulation and the engineering of hybrid spintronic-photonic devices. Recently, two-dimensional (2D) van der Waals (vdW) anti-ferromagnetic (AFM) semiconductors have provided a promising candidate for studying exciton dynamics, due to the emerging spin-orbit-entangled many-body states[17,18], along with their terahertz resonance frequency[19], ultrafast dynamic response[20,21], and strong anti-interference capability in optimizing the lifetime and stability of excitons[22,23]. In addition, the strong coupling effect between the magnetic order and exciton enables the formation of a magnon-exciton coupling state, which expands the horizon of 2D materials, particularly enticing for the applications that exploit coherent magnons as energy-efficient information carriers[16] in opto-spintronics and magnonics, or as interconnects in hybrid quantum systems[24]. Nevertheless, achieving controllable tunability of excitonic optical properties through magnetic field engineering, so as to address the demands of multi-scenario applications, remains a critical challenge in current photoelectronic technology.

CrSBr is an air-stable vdW layered AFM material with a direct semiconductor bandgap of approximately 1.5 eV[25,26], which thus potentially endows the simultaneous modulation of optical and magnetic behaviors. Correspondingly, some pioneering works[27–31] have revealed the tunability of electron transport properties and exciton

energies within the magnetic phase. These observations suggest a strong relevance between the magnons and excitons in a single CrSBr system, rendering it ideal for exploring the coherent coupling state. Note that the coupling of Wannier excitons to magnetic order in CrSBr comes from the spin-dependent electron-exchange interactions[32,33]. By applying an external magnetic field, one can foresee the identical coherent behavior that a spin wave may coherently modulate the electronic structure, which is reflected in the dominant excitonic transitions in a 2D semiconductor. The altering excitonic photoluminescence (PL) under spin order modulation of the magnetic field in CrSBr validates the feasibility of tailoring interlayer electronic and exciton effects by manipulating magnetic order[34,35]. However, to date, such manipulation is confined solely to a single CrSBr system, which not only demands the complicated application of gate voltage[36,37] and external magnetic field apparatus[38,39], but is also intrinsically dominated by the emergence of ferromagnetic (FM) order, thereby disregarding the unique AFM advantages inherent to CrSBr[40–42]. More critically, existing methodologies are limited exclusively to unidirectional modulation towards reducing exciton energy[33–35,38], and have not yet achieved bidirectional reversible modulation. Note that the pseudospin induced by the magnetic proximity interactions can provide a significantly higher interfacial coupling field than the external effective magnetic field[43,44], while few studies have focused on exploring this strategy to manipulate the exciton emission of CrSBr.

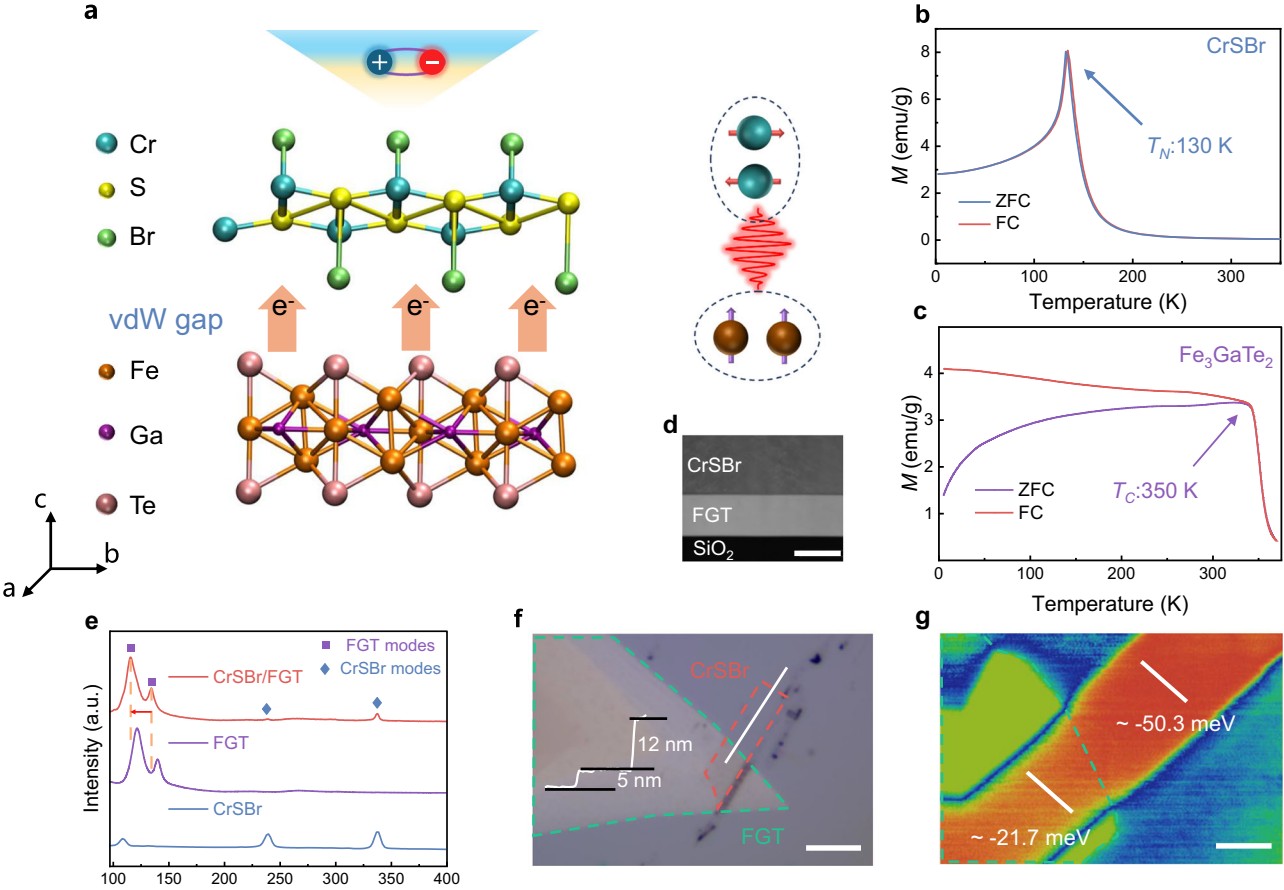

**Fig. 1 | Schematic illustration and physical properties of CrSBr/FGT heterostructures. a** Schematic diagram of CrSBr/FGT heterostructures and the magnetic coupling effect between different spins. **b**, **c** M-T curves of CrSBr (**b**) and FGT (**c**) bulk crystals, measured in a 0.1 T external field along the out-of-plane. **d** TEM image of the cross-sectional region of the heterostructure. Scale bar, 50 nm. **e** Raman spectra of CrSBr flake, FGT flake, and FGT/CrSBr heterostructure. The square symbols represent the Raman signals of FGT, while the diamond symbols stand for the phonon modes of CrSBr. **f** The optical micrograph and thickness line profile of FGT/CrSBr heterostructures on SiO₂/Si substrate. The red and green dashed lines sketch the areas of CrSBr and FGT, respectively. Scale bar, 20 μm. **g** KPFM topography and work function plots of FGT/CrSBr heterostructures. The red area is pristine CrSBr, and the green dashed line marks the area of FGT. Scale bar, 5 μm.

This hinders the opportunities to unlock more possibilities for tuning its optical and electronic properties, limiting the advancement of CrSBr-based devices and applications. Moreover, such AFM/FM heterostructure architectural design potentially serves as a fundamental building block for next-generation excitonic devices. Our work extends this field into new territory by constructing and exploiting magnetic exchange interactions in a vdW magnetic heterostructure.

Here, we report interfacial magnetic interaction-modified spin-exciton coupling in a 2D magnetic heterostructure formed by CrSBr and $Fe_3GaTe_2$(FGT). The features of exciton PL in CrSBr exhibit a tight dependence on the magnetic transition temperatures of both CrSBr and FGT, suggesting the existence of spin-exciton coupling. In contrast to the common PL redshift in CrSBr, we demonstrate that the exciton energy exhibits an anomalous blueshift, originating from the suppression of interlayer electron-hole recombination induced by interfacial magnetic interaction. Through magnetotransport tests, we demonstrate that the magnetic properties of both FGT and CrSBr are modulated by the interfacial interaction. Furthermore, in combination with microscopic-level synchrotron X-ray photoemission electron microscopy (XPEEM), we reveal an orbital-mediated charge transfer mechanism (from Fe $e_g$-orbital to Cr $t_{2g}$-orbital) for the interfacial magnetic interaction between CrSBr and FGT, which alters the magnetic domain of CrSBr. Micromagnetic simulations combined with theoretical calculations further support that the observed excitonic blueshift originates from the suppression of interlayer electron-hole recombination driven by interfacial magnetic interactions. Our work provides an approach for designing coherent behavior between spin and exciton, offering new insights into the potential application of AFM semiconductors, exemplified by CrSBr, in the field of opto-spintronics.

## Results and discussion

### Structure and physical properties of CrSBr/FGT heterostructure

We synthesized high-quality CrSBr and FGT single crystals (Figs. S1–S4) and fabricated the CrSBr/FGT heterostructure by using few-layer flakes exfoliated from their bulk. Figure 1a presents a schematic illustration of the stacking structure and incorporation coupling between different spin orders in the heterostructure. To explore spin-related coupling behavior, we first characterized the magnetic properties of CrSBr and FGT, as shown in Fig. 1b, c, respectively. According to temperature-dependent magnetization ($M-T$) tests, CrSBr exhibits a typical AFM nature with the Néel temperature $T_N = 130$ K, while FGT, on the other hand, exhibits robust ferromagnetism beyond room temperature. A much higher Curie temperature ($T_C = 350$ K) of FGT allows it to constantly provide a magnetic proximity coupling effect throughout the magnetic transition process of CrSBr. The Transmission Electron Microscope (TEM) cross-sectional image (Fig. 1d) confirms the high-quality interface of the heterostructure. Then, we perform Raman measurement to validate the features of the CrSBr/FGT heterostructure, as shown in Fig. 1e. The emergence of hybrid Raman modes from both CrSBr and FGT indicates the specific characteristics of the heterostructure. The optical micrograph further provides images of the CrSBr/FGT heterostructure, as well as cross-sectional analysis (Fig. 1f). The step height of the thickness line profile suggests that CrSBr and FGT are 5 and 12 nm, respectively. Given the differences in conductivity properties, the interface between CrSBr and FGT inevitably involves electronic dynamic processes. Kelvin-probe force microscopy (KPFM) provides the spatial distributions of the surface potential in the heterostructure. As exhibited in Fig. 1g, a significant potential drop of the CrSBr in the heterostructure (~ −21.7 mV), compared to the pristine one (~ −50.3 mV), can be observed from the surface potential in KPFM images. This indicates that FGT acts on inducing a decreased work function and an elevated Fermi level in CrSBr, as a result of charge injection. The specific charge transfer should involve participation in interfacial magnetic interactions and associated coupling phenomena, which will be further discussed later.

### Detection of spin-exciton coupling

Next, we explore the photon properties of CrSBr assisted by ferromagnetic FGT. Figure 2a, b displays the contour mapping diagram for pristine CrSBr and CrSBr/FGT in the temperature range from 80 to 350 K. There is an obvious turning point in the emission energy around the $T_N$ (130 K) in CrSBr/FGT, while pristine CrSBr exhibits an almost linear variation across the entire temperature range. Furthermore, we observe that the temperature-dependent PL spectra exhibit higher photon energy (blueshift) below the $T_N$ for the CrSBr on FGT, compared to pristine CrSBr (Fig. 2c), suggesting a correlation with its AFM phase transition under the action of FGT. Such behavior of excitons modulated by magnetic ordering is recognized as the coupling between magnetic order and excitons. More critically, we perform a fitting analysis of the exciton peak shift using the power-law[45] dependence of the AFM transition and yield a well-fitted result (Fig. S5), which further provides compelling evidence for spin-exciton coupling in the heterostructure. While starting from 130 K, these two kinds of PL peaks are well overlapped during the subsequent warming process, except for several abnormal cases. Within these temperature ranges, CrSBr/FGT exhibits even lower photon energies (redshift) compared to pristine CrSBr, achieving a maximum difference of around 200 K. This observation indicates that the excitonic response in the heterostructure remains sensitive to interfacial magnetic effects over an expanded temperature range, far exceeding the $T_N$ of CrSBr. This suggests that the magnetic proximity effect and residual short-range magnetic correlations[25,26,30] induced by the FGT layer persist well above the intrinsic magnetic transition temperature of CrSBr.

We intuitively demonstrate the difference in PL peaks and the trend of photon energy variation with temperature in Fig. 2d, e, respectively. Compared with pristine CrSBr, a step height greater than 14 meV in photon energy is observed across the critical temperature ($T_N$) in the heterostructure, with a bidirectional modulation magnitude of approximately 6.1% (blueshift) and 8.6% (redshift) relative to the total excitonic linewidth of ~100 meV (Fig. S6). This indicates a high tuning efficiency of exciton energy in CrSBr, achieved through spin-exciton coupling. The unusual redshift of the PL exciton energy in localized regions above $T_N$ is more reasonably attributed to short-range magnetic correlations in CrSBr induced by the underlying FGT. To clarify the specific role of charge transfer in this process, we perform the test of ultraviolet photoelectron spectroscopy (UPS) in Fig. S7, which shows that the Fermi level of FGT is higher than that of CrSBr. Therefore, upon contact, when the Fermi levels align for both FGT and CrSBr, the energy level of their conduction bands would also match up, leading to charge redistribution at the interface. Furthermore, we fabricated a heterostructure by stacking the nonmagnetic metallic material $WTe_2$ (whose conductivity is comparable to that of FGT[46,47]) with CrSBr. The KPFM results (Fig. S8a) demonstrated that charge transfer still occurs between the two materials in the same direction. However, in the absence of interfacial magnetic interactions, the PL spectra (Fig. S8b, S8c) exhibit no correlation with the magnetic order of CrSBr. This further confirms that the exciton energy changes in CrSBr heterostructure cannot only be fully explained by charge transfer, but also are primarily attributed to interfacial magnetic interaction-modified spin-exciton coupling. The detailed analysis of the elevated photon energy in CrSBr/FGT, resulting from spin-exciton coupling, will be elaborated thereafter.

### Altered magnetic properties by the interfacial interaction

To further probe the effect of the interfacial magnetic interactions on spin ordering and magnetic properties of CrSBr, we conducted

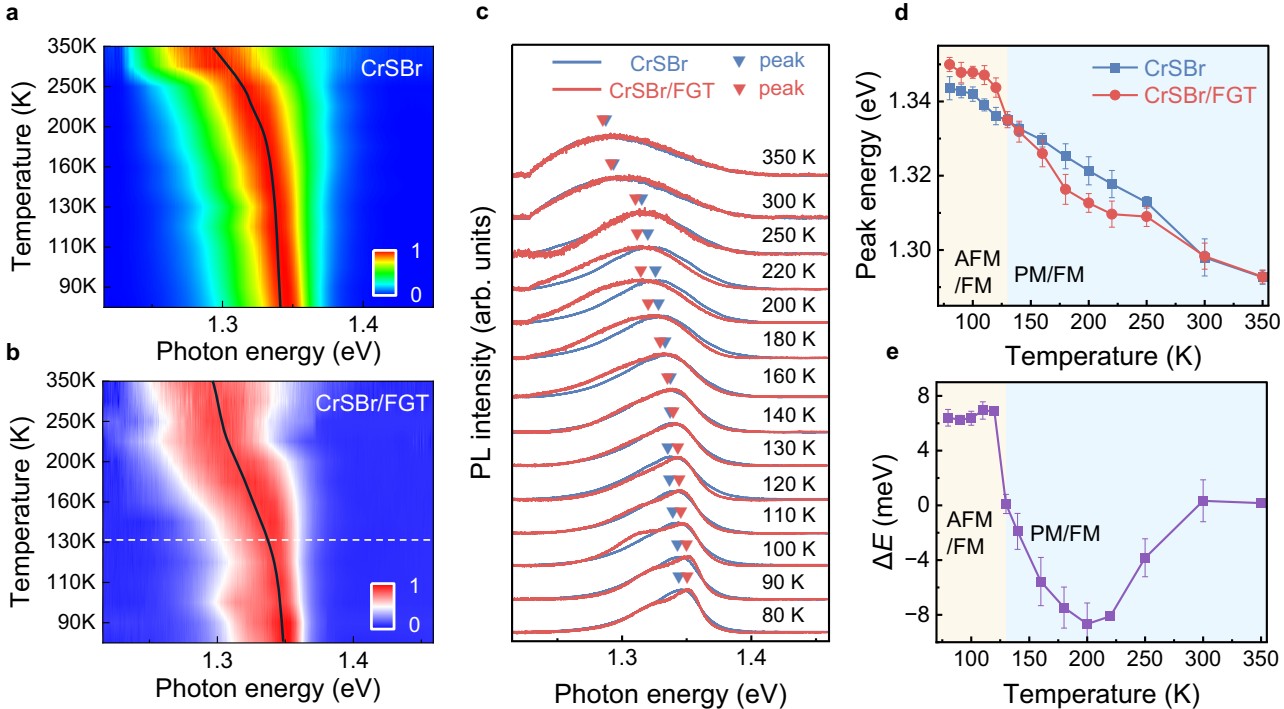

**Fig. 2 | Photon properties of pristine CrSBr and CrSBr/FGT. a, b** Contour color map of the normalized PL intensity with varying temperatures for pristine CrSBr (**a**) and CrSBr/FGT (**b**). **c** Temperature-dependent PL spectra for pristine CrSBr and CrSBr/FGT. The triangles correspond to the energy position of their PL peaks. **d** Variations of the PL peak positions for pristine CrSBr and CrSBr/FGT under different temperatures. Error bars represent the standard deviation of three measurements. **e** The difference of PL peak shift between CrSBr and CrSBr/FGT as a function of temperature. Error bars represent the standard deviation of three measurements.

magnetotransport tests using standard Hall bars. Figure 3a gives the schematic diagram of the fabricated device that FGT stacks onto the CrSBr, establishing direct contact between the electrode and the CrSBr. Figure 3b shows the optical photograph of the pristine CrSBr and FGT/CrSBr heterostructure device (Fig. S9). The spin orientation correlating with the interlayer tunneling determines the behavior of resistance ($R$) under the magnetic field, making magnetoresistance a sensitive probe to capture the spin ordering dynamics. In Fig. 3c, the temperature-dependent magnetoresistance of pristine CrSBr exhibits a correlation with the applied field, showing a specific spin-flip transition below $T_N$. The spin-flip field ($H_s$), corresponding to the reorientation of spin order along the direction of fields, gradually increases as the temperature decreases. Within a low magnetic field, the spins of CrSBr interlayers antialign along the easy $b$-axis, leading to a high tunneling resistance. When the $H_s$ is achieved as fully polarized spins point along the direction of the magnetic field, the uniformly aligned spins facilitate the process of tunneling. For the FGT/CrSBr heterostructure device (Fig. 3d), $H_s$ is enhanced. This indicates that the spin orientation along the easy axis in CrSBr has been strengthened, thereby necessitating higher magnetic fields to force spin reorientation. We further identify that this strengthened spin orientation shows a strong positive correlation with the exciton energy modulation capability (Fig. S10). Besides, we also extracted the $H_s$ and magnetoresistance ratio (MRR) from CrSBr and FGT/CrSBr devices, and plotted them as a function of temperature, as shown in Fig. 3e, f. Overall, the $H_s$ of FGT/CrSBr are significantly larger below $T_N$, with even more pronounced differences at low temperatures. However, the MRR of FGT/CrSBr exhibits an anomalous increase trend with the temperature below $T_N$, which can be attributed to the interfacial magnetic coupling in CrSBr that stabilizes the spin orientation, leading to a higher magnetoresistance. This enhanced magnetoresistance may also influence the interlayer electronic interactions, potentially affecting the excitonic behavior.

In addition, we fabricated another heterostructure in which CrSBr stacks onto the FGT (Fig. S11) to explore the impact of magnetic coupling on the FGT magnetism. To clarify the signal origin, we independently measured the Hall response of CrSBr and found a negligible Hall resistance (Fig. S12), demonstrating that the Hall signals observed in the heterostructure originate solely from the FGT layer. Figure 3g presents the illustrative diagram of the Hall devices for CrSBr/FGT heterostructure, and both the optical images of pristine FGT and CrSBr/FGT are given in Fig. 3h. The Hall resistance $R_{xy}$ of pristine FGT, measured under an out-of-plane field, exhibits regular square-shaped hysteresis loops in the whole temperature range from 5 K to 300 K (Fig. 3i), suggesting robust ferromagnetism that agrees with its single-crystal magnetic test results (Figs. S13, S14). With increasing temperature, the $R_{xy}$ and coercivity field ($H_c$) gradually drop due to the thermal fluctuations. However, Fig. 3j shows a distinct evolution of $R_{xy}$ and $H_c$ for FGT in the heterostructure when interfacial magnetic coupling exists. By further extracting these data and plotting their variation with temperature, we found that the $H_c$ of CrSBr/FGT is significantly larger than that of the pristine FGT below the temperature close to $T_N$ (Fig. 3k). Additionally, under the interfacial magnetic interaction, the $R_{xy}$ of CrSBr/FGT no longer exhibits a monotonically decreasing behavior with temperature increasing, but shows a trend of increasing to around $T_N$ and then decreasing (Fig. 3l). It is evident that within the temperature range of magnetic order in CrSBr, the magnetism of FGT does indeed enhance. All these results declare that the interfacial magnetic interaction in the CrSBr/FGT heterostructure is reciprocal, which affects the magnetic properties of both CrSBr and FGT.

## Interfacial magnetic interaction in the heterostructure

Given the intensive impact of magnetic coupling on spin-correlated phenomena, we further precisely investigate the features of the interfacial magnetic interaction. Element-resolved XPEEM enables

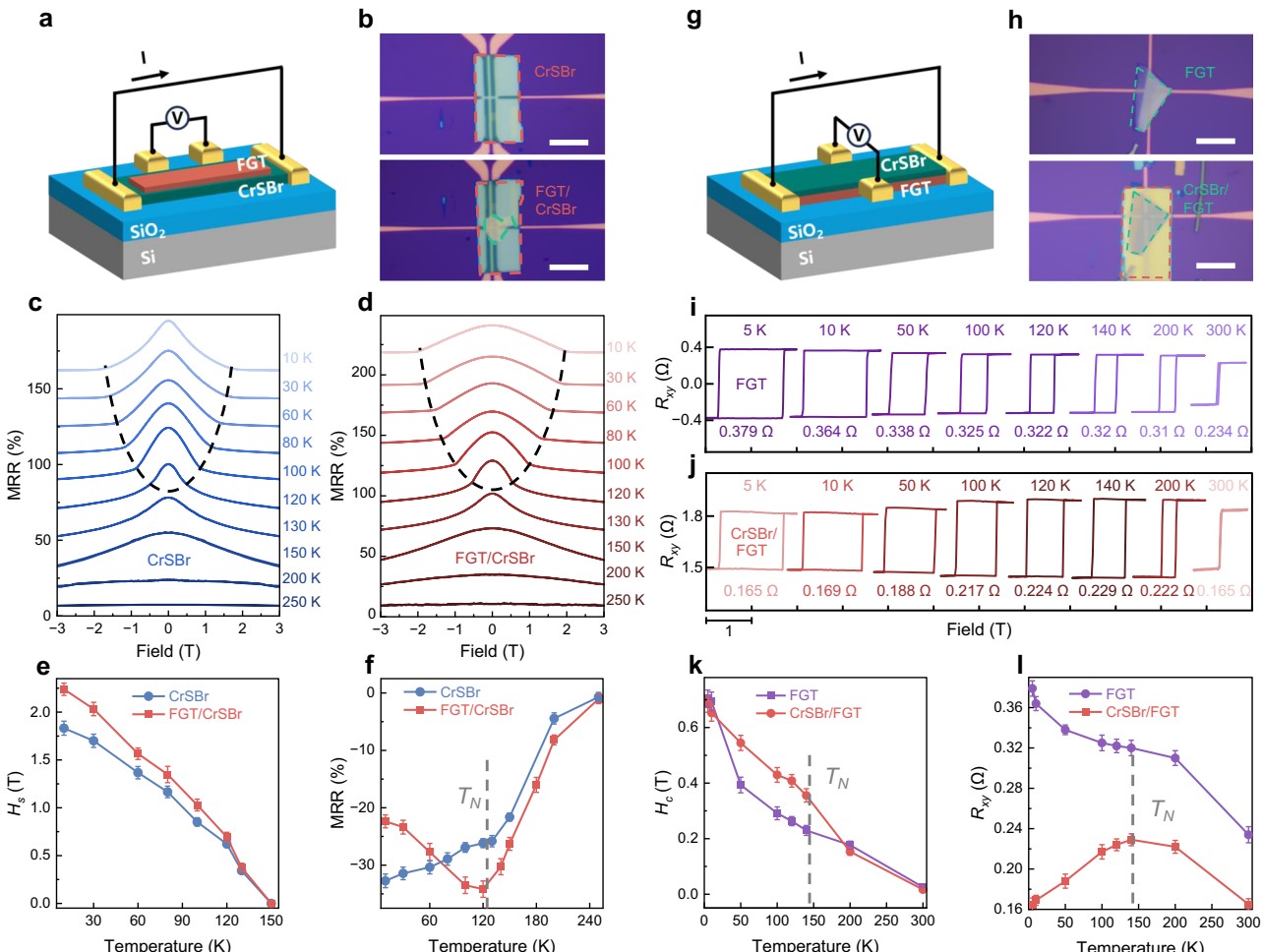

**Fig. 3 | Magnetotransport and Hall effect measurements. a** Schematic diagram of the FGT/CrSBr device structure. FGT is placed on the top layer, and CrSBr is positioned on the bottom layer to make direct contact with the electrodes. **b** Optical images of the CrSBr device (upper panel) and the FGT/CrSBr device (lower panel). The red and green dashed lines indicate the areas of CrSBr and FGT, respectively. Scale bar, 20 μm. **c, d** MRR of the pristine CrSBr (**c**) and the CrSBr in the heterostructure (**d**) versus magnetic field at various temperatures, measured with the field oriented along the out-of-plane. MRR defined as $[(R_H - R_{H=0})/R_{H=0}] \times 100\%$. The dashed line delineates the onset field of the spin flip. **e, f** The extracted $H_s$ (**e**) and MRR (**f**) vary with temperature. Error bars represent the standard deviation of three measurements. **g** Schematic diagram of CrSBr/FGT Hall device structure. CrSBr is placed on the top layer, and FGT is positioned on the bottom layer. **h** Optical images of the FGT device (upper panel) and the CrSBr/FGT device (lower panel). The red and green dashed lines outline the areas of CrSBr and FGT, respectively. Scale bar, 20 μm. **i, j** $R_{xy}$ of the pristine FGT (**i**) and the FGT in the heterostructure (**j**) versus magnetic field at various temperatures, measured with a perpendicular applied magnetic field. The numerical values beneath the curves correspond to the $R_{xy}$ parameters extracted from individual traces. **k, l** The extracted $H_c$ (**k**) and $R_{xy}$ (**l**) as a function of temperature. Error bars represent the standard deviation of three measurements.

ultrahigh-spatial resolution imaging to elucidate the magnetic domain structure, magnetization direction, and interlayer interactions. All XPEEM data were recorded at 100 K after zero-field cooling with normal incidence (90°) of X-ray on the sample surface. Exceptionally, Fig. 4i was measured using grazing-incidence X-ray (16°). Figure 4a gives a schematic diagram of the principle of XPEEM. According to the X-ray absorption spectroscopy (XAS) of Fe $L$-edge (Fig. 4b), both the $L_3$ and $L_2$ edges of the heterostructure shift to higher energy compared to pristine FGT, verifying its electron loss[48]. Moreover, the shoulder peak on the high-energy side of the $L_3$ edge becomes more pronounced, indicating that electrons are depleted from the higher-energy, $e_g$, orbitals, leading to an increased population of unoccupied states[49,50]. In Fig. 4c, Cr $L$-edge XAS of the pristine CrSBr exhibits a typical feature of $Cr^{3+}$, whereas in the heterostructure, the shoulder peak on the low-energy side of the $L_3$ edge is weakened, and the fine structure at the $L_2$ edge disappears. These characteristics are indicative of a decrease in valence state after gaining electrons, and the features of this shoulder peak also suggest that electrons are transferred from the Fe $e_g$-orbital

in FGT to the Cr $t_{2g}$-orbital in CrSBr. To further explore the evolution of the magnetic moment under the magnetic coupling, we further conducted X-ray magnetic circular dichroism (XMCD) and X-ray magnetic linear dichroism (XMLD) spectra. We then extract the Fe $L$-edge XMCD data from the areas marked by the red and purple rectangles in the XMCD-PEEM. As shown in Fig. 4d, e, the Fe $L$-edge XMCD signal of pristine FGT (−25.8%), extracted from the bright (rectangle) area in Fig. 4f, is significantly higher than that of the FGT in the CrSBr/FGT heterostructure (−18.5%). We also extract the data from the dark (circle) areas, in which a similar intensity variation of XMCD can be obtained in the opposite polarization direction (Fig. S15). This indicates that when forming a heterostructure with CrSBr, the magnetic moment of FGT is weakened. Subsequently, the XPEEM imaging of the magnetic domain at the Fe $L_3$ edge photon energy is performed and presented in Fig. 4f. It is evident that the pristine FGT displays a typical domain, and even in the heterostructure area with CrSBr, there is no significant change in this magnetic domain phase. We further demonstrate that this magnetic domain can be well maintained up to

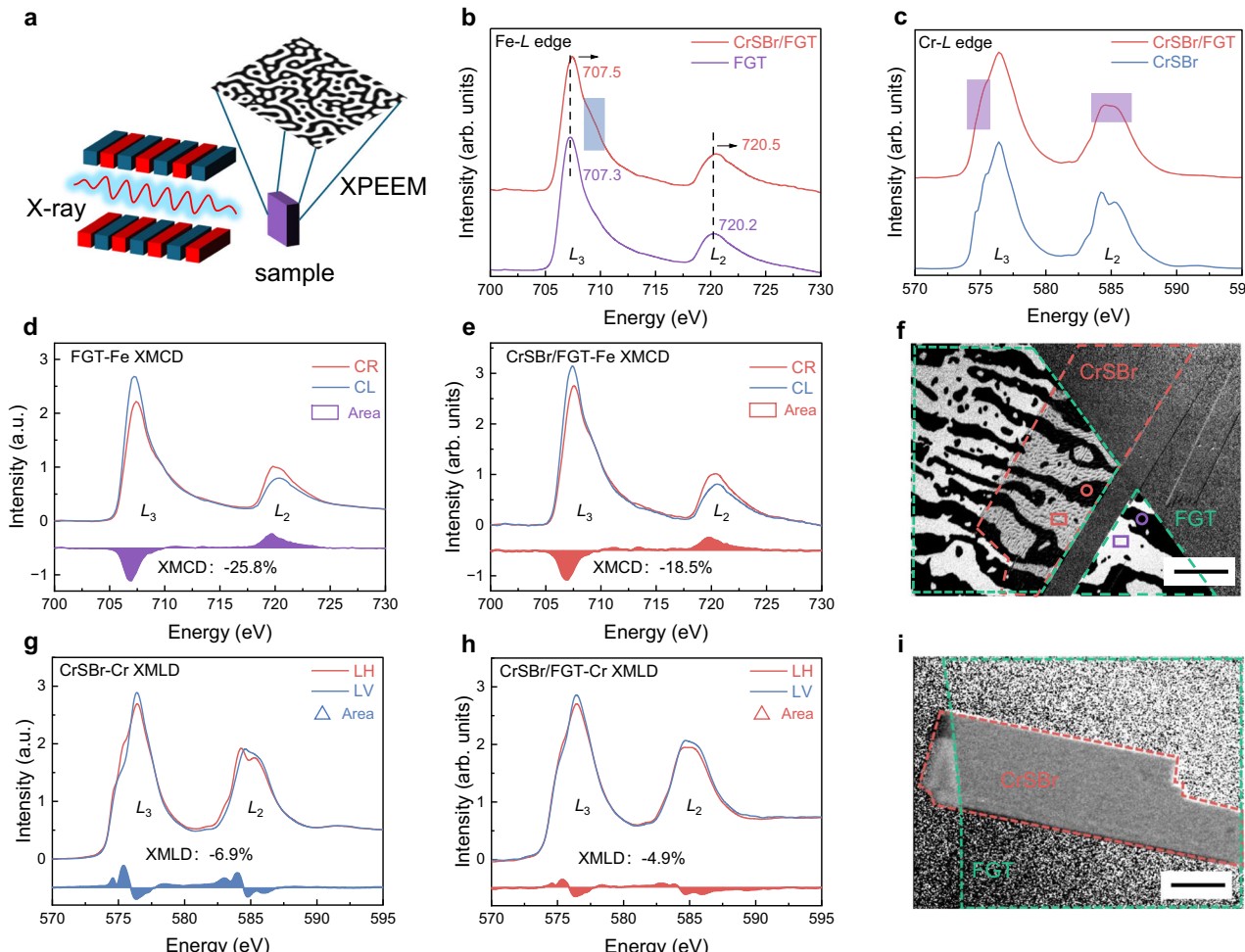

**Fig. 4 | Features of magnetic texture and magnetic moment. a** Schematic diagram of XPEEM measurement. The modulated X-rays are incident on the sample, and magnetic domain imaging is obtained based on the intensity of the photoelectrons. **b** XAS spectra of Fe *L*-edge in FGT and in CrSBr/FGT. **c** XAS spectra of Co *L*-edge in CrSBr and in CrSBr/FGT. **d, e** The extracted XMCD of Fe-*L* edge in FGT (**d**) and in CrSBr/FGT (**e**). CR: right-circular polarized X-ray. CL: left-circular polarized X-ray. **f** XMCD-PEEM images of Fe *L*-edge with the incident X-ray perpendicular to the sample surface. The red and green dashed lines mark the areas of CrSBr and

FGT, respectively. The rectangular and circular frames, respectively, represent the marked areas used for Fe XMCD data extraction. Scale bar, 4 μm. **g, h** The extracted XMLD of Cr *L*-edge in CrSBr (**g**) and in CrSBr/FGT (**h**) from Cr XMLD-PEEM. LH: horizontal-linear polarized X-ray. LV: vertical-linear polarized X-ray. The inserted triangles stand for the marked areas used for Cr XMLD data extraction. **i** XMCD-PEEM images of Cr *L*-edge with a 16° X-ray incidence angle relative to the sample surface. The red and green dashed lines mark the areas of CrSBr and FGT, respectively. Scale bar, 4 μm.

room temperature (Fig. S16), indicating the stability of the FGT magnetic structure.

Besides, the extracted Cr XMLD signal from the XMLD-PEEM image (Fig. S17) also undergoes suppression, as it decreases from −6.9% to −4.9% (Fig. 4g, h), suggesting a reduced AFM spin moment. To directly probe the in-plane magnetic domains of CrSBr and to validate the interfacial interaction in the heterostructure, we further performed Cr *L*-edge XMCD-PEEM measurements under grazing-incidence geometry (16°) (Fig. 4i), which enhances sensitivity to the in-plane magnetic component of Cr moments. Notably, compared with the multidomain state of pristine CrSBr, CrSBr in the heterostructure exhibits a magnetic single-domain with decreased magnetic contrast. This direct observation clearly indicates that the enhanced magnetic domain wall energy and decreased magnetic moment for CrSBr in the CrSBr/FGT heterostructure further demonstrating the presence of interfacial magnetic interaction between CrSBr and FGT. In contrast, the Cr *L*-edge XMCD-PEEM acquired under normal-incidence geometry (Fig. S18) shows that the magnetic effect of the heterostructure failed to induce the appearance of observable XMCD signals at the Cr *L₃* edge photon energy, which demonstrates that there is no spin

canting along the out-of-plane. Furthermore, according to the Cr *L*-edge XMLD-PEEM image (Fig. S17), we further verify that the in-plane orientation of the Néel order in both pristine CrSBr and CrSBr/FGT remains unchanged, maintaining stability along the original AFM *b*-axis. The absence of Néel order canting or flopping in CrSBr indicates that the observed reduction in XMLD intensity originates from a diminished magnetic moment of CrSBr. Based on the above discussion, although the interfacial magnetic interaction in the CrSBr/FGT heterostructure changes the magnetic domain of CrSBr, it does not alter the intrinsic AFM spin orientation. On the other hand, regarding the observed reduction in magnetic moments for both CrSBr and FGT within the heterostructure, an orbital-mediated electron transfer mechanism should be considered, which affects the unpaired electrons and net moments (Fig. S19). Specifically, in CrSBr, where $Cr^{3+}$ ions possess three unpaired electrons in their outermost *d*-orbitals (each occupying a distinct $t_{2g}$ orbital), the introduction of extrinsic electrons through interfacial charge transfer enables pairing with these localized spins. This electron population in the $t_{2g}$ consequently reduces the net unpaired electron count, leading to diminished magnetic moments. For the FGT subsystem, where both $Fe^{2+}$ and $Fe^{3+}$ ions maintain

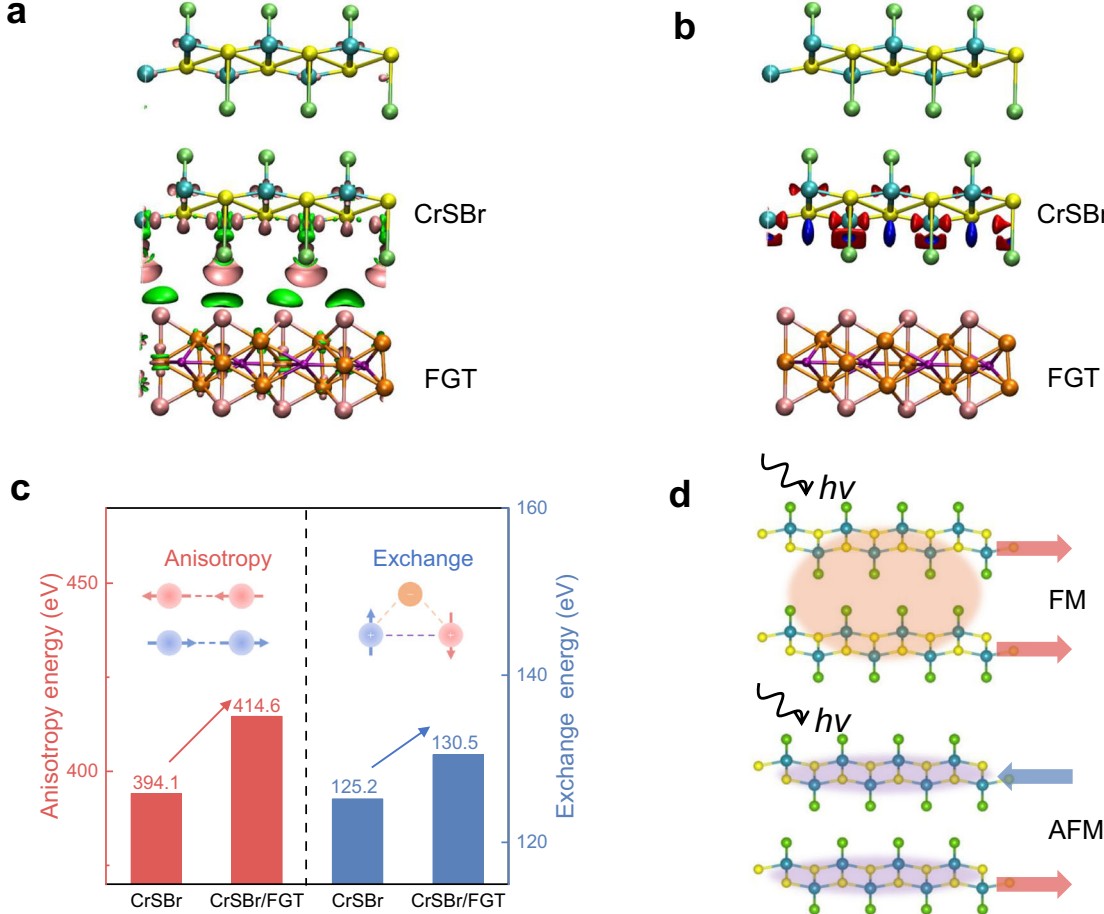

**Fig. 5 | Mechanism of the anomalous exciton blueshift. a** The charge density differences for CrSBr/FGT heterostructure. The pink and green areas depict charge accumulation and depletion, respectively. **b** The spin polarization rate difference ($\Delta P$) of CrSBr in the heterostructure. The red and blue areas represent positive and negative values of the spin polarization rate, respectively. **c** Micromagnetic simulation results of the exchange interaction and magnetic anisotropy energies of CrSBr in the pristine and heterostructure (extracted from each CrSBr layer). **d** Schematic illustration of the exciton recombination process in CrSBr under the enhanced AFM configuration.

unpaired electrons in their $e_g$ orbitals, the loss of any single electron from these degenerate orbitals thus results in a quantized decrease in spin-polarized electrons and a weakened magnetic moment. This orbital-mediated electron transfer mechanism is well supported by our XAS data (Fig. 4b, c), and the interfacial charge transfer can simultaneously modify the electronic structure of Cr, thereby exerting a profound impact on macroscopic magnetic properties. Specifically, the increased electron occupancy in Cr-3$d$ orbitals of CrSBr enhances their electron cloud density and amplifies the spatial overlap of adjacent Cr atomic orbitals[51–53], which in turn strengthens orbital hybridization, stabilizes the spin configuration, and reinforces the interfacial magnetic interaction.

## Modulation mechanism of spin-exciton coupling by interfacial magnetic interaction

Lastly, we discuss the anomalous increase of exciton energy in CrSBr/FGT below $T_N$. To understand this phenomenon from a microscopic perspective, we performed a series of theoretical calculations and simulations. Density functional theory (DFT) calculations reveal a pronounced redistribution of charge density in the CrSBr/FGT heterostructure (Fig. 5a). Quantitative charge analysis (Table S1, Supporting Information) shows a net electron transfer of approximately 0.007 e/Å² from FGT to CrSBr, in agreement with our KPFM and XAS measurements, further confirming the existence of interfacial charge

transfer. Notably, such charge transfer occurring between magnetic materials provides a crucial electronic-structure foundation for the establishment of interfacial magnetic interaction. On this basis, we further extract the spin polarization rate difference ($\Delta P$) of CrSBr in the heterostructure from the DFT calculations (Fig. 5b), which can be expressed as $\Delta P = |(N_{up}^{hetero} - N_{down}^{hetero})/N_{total}^{hetero}| - |(N_{up}^{CrSBr} - N_{down}^{CrSBr})/N_{total}^{CrSBr}|$, where $N_{up}$, $N_{down}$ and $N_{tot}$ denote the numbers of spin-up, spin-down and total electrons, respectively. The superscripts "hetero" and "CrSBr" refer to the CrSBr in the CrSBr/FGT heterostructure and pristine CrSBr, respectively. A positive $\Delta P$ indicates that the spin-polarized rate of CrSBr is enhanced in the heterostructure compared to pristine CrSBr, providing direct microscopic electronic-structure evidence for the modulated spin configuration of CrSBr. To further elucidate how these electronic-structure modifications influence the magnetic properties, we performed micromagnetic simulations based on parameters extracted from both experiments and theoretical calculations. Micromagnetic simulations based on MuMax3[54] serve as a powerful tool for capturing the evolution of magnetic energies and spin configurations in magnetic materials, particularly for our constructed heterostructure model incorporating interlayer magnetic coupling between FGT and CrSBr (Fig. S20). The simulation results demonstrate that both the magnetic anisotropy energy and the exchange interaction energy of CrSBr are enhanced in the heterostructure (Fig. 5c), which is consistent with the emergence of

a magnetic single-domain of CrSBr observed in the XMCD-PEEM measurement. This elevated magnetic energy landscape effectively stabilizes the AFM spin configuration of CrSBr, which is in excellent agreement with the robust spin orientation in XMLD-PEEM and the increased spin flip field in transport measurements, and is further supported by the enhanced spin polarization rate revealed by the DFT calculations.

Combining the above experimental observations, micromagnetic simulations, and DFT calculations, we establish the following physical picture: interfacial charge-transfer-driven magnetic coupling enhances the AFM ordering of CrSBr, manifested by the concurrent enhancement of its exchange interaction and magnetic anisotropy. It is worth noting that for A-type AFM CrSBr, antiparallel spin alignment between adjacent layers strongly suppresses interlayer electron-hole recombination[35]. When an external magnetic field or other perturbations drive the interlayer spins into a ferromagnetically aligned configuration, spin-allowed interlayer charge transfer channels are opened, leading to enhanced interlayer electron-hole hybridization and a redshift of the exciton energy[35,55]. In contrast, in our CrSBr/FGT heterostructure, the interfacial magnetic interaction further strengthens the AFM order of CrSBr, thereby more effectively suppressing interlayer electron-hole recombination and significantly reducing its contribution to optical transitions (Fig. 5d). Our work ultimately gives rise to an anomalous blueshift of the exciton emission energy below $T_N$ through interfacial magnetic interaction. When the temperature exceeds the $T_N$ of CrSBr, its AFM order gradually diminishes, and the interfacial magnetic interaction-modulated ability correspondingly weakens, resulting in an abrupt change in the PL energy across the $T_N$ (Fig. 2d). In the temperature range above the $T_N$ of CrSBr but below the $T_C$ of FGT, ferromagnetically ordered FGT may still locally induce FM correlations in CrSBr, which can reopen interlayer electronic hybridization channels and lead to a redshift of the exciton energy. As the temperature is further increased beyond the $T_C$ of FGT (above 300 K), this induced effect vanishes, and the exciton energy in the CrSBr/FGT heterostructure gradually converges to that of pristine CrSBr. This tunable exciton energy between blueshift and redshift, driven by interfacial magnetic interaction-dependent spin-exciton coupling, may unlock the potential for arbitrary manipulation of exciton energies to enable precise control over the quantum states of excitons in quantum information storage.

In summary, we demonstrate the bidirectional tunability of exciton energy in a 2D CrSBr/FGT magnetic heterostructure, governed by spin-exciton coupling modified through interfacial magnetic interactions. The FM order in FGT modulates the spin configuration of CrSBr, which is consistently reflected in the evolution of magnetic-sensitive exciton energy, leading to an anomalous photon emission energy blueshift below $T_N$. These results establish a direct correlation between interfacial spin states and exciton energy, highlighting spin-mediated control of excitonic properties in vdW heterostructures. Based on experimental measurements and theoretical simulations, we demonstrate that the interfacial charge-transfer-driven magnetic coupling in the heterostructure enhances both the magnetic anisotropy and exchange interaction of CrSBr, effectively stabilizing its AFM spin configuration, which suppresses interlayer electron-hole recombination and increases the spin-flip field. Notably, the exciton energy shift can be bidirectionally modulated between positive and negative states relative to pristine CrSBr, with the magnitude of blueshift and redshift exceeding 6% and 8% of the total bandwidth, respectively. This bidirectional control highlights significant flexibility in device design and offers a valuable reference for future studies on wavelength control in laser systems. Our findings present an effective strategy to tailor spin-exciton coherence, opening avenues for AFM semiconductors like CrSBr in opto-spintronic devices such as spin-logic gates and non-volatile magneto-optical memory systems.

## Methods

### Sample preparation

Single crystals of CrSBr were successfully synthesized using the chemical vapor transport (CVT) method, with iodine as the transport agent. High-purity chromium (99.9%), bromine (99.99%), and sulfur (99.99%) powders were weighed according to the stoichiometric ratio of 1:1:1, mixed, and sealed in quartz tubes inside an Ar-filled glove box. The sealed quartz tubes were then placed in a two-zone furnace, where the temperature gradient of the source and growth zones was set at 1173 K (hot zone) and 1073 K (cold zone), respectively. After one week of isothermal heating, high-quality CrSBr single crystals were successfully obtained. Single crystals of FGT were also grown by the CVT method. The CVT growth process began with a mixture of iron power (99.95%), gallium pellet (99.99%), and tellurium lump (99.99%) in a stoichiometric molar ratio of 3:1:2, prepared in an Ar-filled glove box. Subsequently, the mixture and an amount of the transport agent iodine were sealed together into an evacuated quartz tube. The sealed quartz tube was then placed in a two-zone tubular furnace. The temperatures were ramped to 1023/923 K over six hours, maintained for one week, and then cooled to room temperature naturally. Finally, high-quality FGT single crystals with a smooth surface and metallic luster were obtained.

### Device fabrication and transport measurement

Pre-patterned Hall bars were fabricated using ultraviolet lithography (Aligner Suss, MABA6) and electron-beam deposition (LAB 18) of Cr/Au (5/20 nm) on a 300-nm-thick $SiO_2$/Si substrate. The high-quality vdW bulk single-crystal CrSBr and FGT were mechanically exfoliated into few-layer flakes inside an Ar-filled glove box. The transport was measured in the PPMS-9T (Quantum Design) system. The anomalous Hall effect and magnetoresistance measurements were performed using a Keithley Model 2400 C source meter in conjunction with a Keithley Model 2612B nanovoltmeter. The devices were annealed at 393 K in an Ar-filled glove box to eliminate potential air gaps and ensure secure contact between the flakes.

### Raman scattering and PL measurement

Raman spectroscopy and PL spectroscopy were carried out on the LabRamHR Evolution laser Raman spectrometer produced by the French JY company. The continuous laser of 532 nm was employed for the Raman and PL measurements; a 50 × objective lens was used to focus the laser beam onto the sample surface, achieving a spot size of approximately 1 μm. The beam power was kept below 2 mW to avoid damaging the sample. For the measurement, we used a 30 s integration time and averaged a total of 10 spectra to increase the signal-to-noise ratio, and subtracted a background spectrum taken on the $SiO_2$/Si substrate under the same conditions. During the assembly of the heterostructure, we designed the configuration to simultaneously obtain pristine CrSBr, pristine FGT, and CrSBr/FGT regions within a single sample, at each temperature point, measurements are taken separately on these three regions to acquire a complete set of temperature-dependent data. The sample was housed in a closed-cycle cryostat, and the Linkam THMS600 and liquid nitrogen were used for cooling the sample for a variable temperature test with 80 K–370 K.

### Structural and magnetic measurements

The X-ray diffraction (XRD) patterns of CrSBr and FGT were obtained by using a Multifunctional Rotating-anode X-ray Diffractometer with Cu-$K\alpha$ radiation ($\lambda = 1.54178$ Å). The Scanning Electron Microscope (SEM) images and Energy Dispersive Spectroscopy (EDS) mapping were captured by Gemini SEM 500. The atomic force microscope (AFM) images and Kelvin-probe force microscopy (KPFM) measurements were taken with a Dimension Icon Atomic Force Microscope produced by Bruker Nano Inc. in tapping mode. The magnetic

properties were measured by a commercial superconducting quantum interference device (SQUID) magnetometer from Quantum Design. The XPEEM measurements were conducted at the MAXPEEM beamline of the MAX IV Laboratory (normal incidence of X-ray) and the I06 beamline at Diamond Light Source (16° incidence of X-ray). Fe and Cr XMCD-PEEM raw images were recorded by 1 s exposure time with right (R) and left (L) circularly polarized light at Fe $L_3$ resonance at 707.3 eV and Cr $L_3$ resonance at 576.4 eV. The XMCD asymmetry is given by $(I^L - I^R)/(I^L + I^R)$. Cr XMLD-PEEM raw images were recorded by 5 s exposure time with linear horizontal (LH) and vertical (LV) polarized light at 575.4 eV, and the XMLD asymmetry is given by $(I^{LH} - I^{LV})/(I^{LH} + I^{LV})$. The UPS data were collected at the ARPES beamline (BL13U) of the National Synchrotron Radiation Laboratory (NSRL, China), with a DA30 analyzer and 91.5 eV photon energy.

## Micromagnetic simulations

For the micromagnetic simulations performed in MuMax3, we used a cell size of 2 nm × 2 nm × 1 nm and a total mesh of 512 × 512 × 4 cells (corresponding to 2 layers of FGT and 2 layers of CrSBr). Procedure for micromagnetic parameter extraction: first, the interlayer coupling in the heterostructure is turned off (i.e., FGT and CrSBr are treated as independent regions), after which the anisotropy and exchange energies are extracted for each layer. Then, the interlayer coupling is enabled, and the anisotropy and exchange energies are re-evaluated for comparison. The exchange stiffness of $A_{ex}$ for FGT and CrSBr were set to 6 pJ/m and 8 pJ/m, respectively, and the saturation magnetizations of $M_s$ were set to $2.4 \times 10^5$ A/m and $1.67 \times 10^5$ A/m, respectively. The Gilbert damping constants α were chosen as 0.02 for FGT and 0.1 for CrSBr. The uniaxial anisotropy constant of FGT was set to $4.8 \times 10^5$ J/m$^3$, with the easy axis along the $z$-direction (out-of-plane), whereas that of CrSBr was set to $1.27 \times 10^5$ J/m$^3$, with the easy axis along the $y$-direction (in-plane). All simulated magnetic states were obtained by relaxing from a random initial spin configuration to reach equilibrium.

## Theoretical calculations

First-principles calculations were performed using spin-polarized density functional theory (DFT) as implemented in the Quantum Espresso package. The electron-ion interactions were treated with the projected augmented wave (PAW) method, and the exchange-correlation effects were described using the generalized gradient approximation (GGA) in the Perdew-Burke-Ernzerhof (PBE) form. A plane-wave kinetic energy cutoff of 85 Ry was employed. The Hubbard-U correction was applied with an effective $U$ value ($U_{eff}$) of 1.0 eV for Fe atoms and ±5.0 eV for Cr atoms. Van der Waals interactions were accounted for using the DFT-D3 scheme. The Brillouin zone was sampled with a 7 × 3 × 1 k-point mesh. Convergence thresholds were set to $10^{-7}$ eV per atom for the self-consistent field iterations and 0.1 eV Å$^{-1}$ for atomic forces during structural relaxation. LBFGS algorithm was used to perform structural optimizations and cell relaxations.

## Data availability

The data generated in this study are provided in the Supplementary Information/Source Data file. Source data are provided with this paper.

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

## Acknowledgments

This work was financially supported by the Strategic Priority Research Program of the Chinese Academy of Sciences (XDA0410401 to W.Y.), the National Natural Science Foundation of China (Grants No. 12105286 to H.D., 12275271 to C.W., 12305369 and 12404129 to C.L.). This work was partially carried out at the USTC Center for Micro and Nanoscale Research and Fabrication, and the Instruments Center for Physical Science, USTC. We acknowledge the beamline BL09U at Shanghai Synchrotron Radiation Facility (SSRF), beamlines MCD-A and MCD-B (Soochow Beamline for Energy Materials), and ARPES Endstations at NSRL for the synchrotron beamtime. The authors acknowledge MAX IV Laboratory for time on Beamline MAXPEEM under Proposal [20230756]. Research conducted at MAX IV, a Swedish national user facility, is supported by the Swedish Research Council under contract 2018-07152, the Swedish Governmental Agency for Innovation Systems under contract 2018-04969, and Formas under contract 2019-02496. We also thank Diamond Light Source for beamtime on beamline I06.

## Author contributions

C.L., H.D., and W.Y. conceived the experiments and supervised the project. H.D., J.L., F.M., and Y.N. performed the XMCD/XMLD-PEEM tests. W.L. and C.L. prepared the sample, performed the characterization, and wrote the manuscript with H.D., Y.F., and C.W. performed the theoretical calculations. R.L., Y.C., and L.C. helped to conduct the heterostructure fabrication. M.F. helped with the Raman/PL tests. H.W. helped to perform the TEM tests. Z.Z. helped with the micro- and nanofabrication. All authors contributed to discussing the results and commenting on the manuscript.

## Competing interests

The authors declare no competing interests.
