## [Transparent Peer Review file · Nature Communications]

Spin-exciton coupling modified by interfacial magnetic interactions in a van der Waals heterostructure

Corresponding Author: Professor Wensheng Yan

Version 0:

Reviewer comments:

Reviewer #1

(Remarks to the Author)

The authors report measurements of the PL spectrum of CrSBr/Fe₃GaTe₂ heterostructures. They observe a strong dependence of the temperature and a steep rise of the main PL peak when moving below the Neel temperature of CrSBr in CrSBr/FGT heterostructures. The results are interpreted in terms of coupling to magnons and phonons. I find the results rather interesting and all experimental data appears to be of high quality. However, I am far from convinced on the general interpretation of the results and cannot recommend the manuscript for publication in Nat. comm.

The authors mention several times that charge transfer occurs between the FGT and CrSBr. Why is that not enough to explain the spin hardening and change in PL spectrum? There is a clear (but) weak coupling between the exciton energies and the magnetic order. But to refer to this as exciton magnon coupling is stretching it in my opinion. The excitons live at completely different energy scales compared to the magnons. Demonstrating dependence of PL on magnetic order is not the same as demonstrating exciton-magnon coupling. Along the same lines: the spin hardening could again be described by charge transfer right? So I cannot see that it is signalling interlayer magnetic interactions. Finally, the phonons are observed to depend on the magnetic order as well. In this case it is perhaps expected to have some degree of magnon-phonon coupling, but I cannot see any explicit evidence of it.

To summarize, I find the results of the manuscript interesting, but not the interpretation and title. The title is difficult to understand for one thing and I do not find any evidence of its claim. What does coherent magnon-exciton coupling mean? Why do you believe this coupling is mediated by phonons? And why do you believe it is boosted by spin-hardening?

Reviewer #2

(Remarks to the Author)

Excitons are of both fundamental interest and technological importance, showing great potential for applications in photospintronic and quantum technologies. In this manuscript, the authors investigate coherent exciton manipulation in a 2D CrSBr/Fe₃GaTe₂ heterostructure by utilizing phonon-mediated magnon-exciton interactions. Through a series of comparative experiments, including temperature-dependent Raman/PL, Hall effect, and XPEEM techniques, they demonstrate that interfacial magnetic coupling and orbital-driven charge transfer induce reciprocal spin hardening. This further leads to phonon-constrained magnon dynamics and a corresponding anomalous increase in exciton energy. These findings hold significant importance for exciton physics and spintronics, which provide a promising approach for modulating excitonic properties in 2D AFM semiconductors and can advance the development of spin-based quantum photonic devices. I believe this work is interesting and suitable for publication in Nature Communications after addressing several concerns:

1. In transport measurements, the FGT/CrSBr heterostructure is regarded as an electrically integrated system. However, it remains unclear whether the observed Hall response exclusively originates from the FGT layer. Given the pivotal role of interface effects in this study, it is important to deconvolute other potential contributions to the Hall signal in the heterostructure. Therefore, the possible influence of CrSBr may be considered. Clarifying this point is critical for a more accurate understanding of the magnetotransport mechanism in the system.
2. Generally, within conventional FM/AFM heterostructures, exchange bias phenomena are frequently observed due to

interfacial spin coupling. It would be informative for the authors to deliberate on whether any such effects exist in the CrSBr/FGT system. This consideration provides a valuable insight into the nature of the FM/AFM interface in this heterostructure.

3. In Figure 3j, the author should explain why the vertical-axis values of the heterostructure do not align with those of the FGT reference in Figure 3i. Moreover, the magnitude of the anomalous Hall resistance in the heterostructure is significantly reduced compared to that of the pristine FGT layer, which also needs a mechanistic explanation for this discrepancy. These clarifications are beneficial to strengthen the interpretation of interlayer interactions and their influence on spin-dependent transport properties.

4. It shows that although the long-range AFM order in CrSBr disappears around 130 K, certain magnetic and optical features persist beyond this temperature. Specifically, the enhanced coercive field in the heterostructure and the increased exciton energy splitting remain observable up to approximately 200 K. This apparent lagging behavior suggests the presence of underlying magnetic correlations or interfacial effects that endure above the Néel temperature. It is advisable for the authors to clarify the possible physical origins of these effects.

5. In Fig. 4i and Fig. S18, the authors should give a detailed discussion on why the XMCD-PEEM images show no discernible signal, and the XMLD-PEEM images reveal a single domain. Additionally, more detailed configuration information should be included in the Methods section regarding how the XMCD and XMLD images were acquired and analyzed.

6. A minor question. It is suggested that error bars should be added in Fig. 3e, 3f, 3k, and 3l to more precisely present the evolution process of the magnetic and magnetotransport properties as the temperature varies.

7. While the paper is highly readable, there are still a few spelling issues. The abbreviation "XMCD-XPEEM" should be corrected to "XMCD-PEEM". The term "flipping" should be more accurately expressed as "flopping" to more appropriately describe the gradual reorientation or tilting of spins under external fields or internal interactions.

In summary, I consider that the paper is interesting and has important guiding implications for the fields of spintronics and optoelectronics, but some minor revisions should be made to clarify the mentioned ambiguities and to meet the quality standards of NC.

Reviewer #3

(Remarks to the Author)

Lan et al present an experimental paper on the excitons of CrSBr layered on Fe₃GaTe₂. They find evidence of phonon coupling to magnetic order, exciton energy shifts with layering, and many other modifications to the CrSBr excitonic behavior. While their experiments are of high quality and contain careful analysis, I have a concern that their claims of tunability are somewhat overstated. Reading the abstract and conclusion, one would expect that they have found a way to control exciton energy over a wide range of energies--when in reality the shift is quite modest.

I have very few technical comments on the manuscript as I am not an expert in them, but I do have the following recommendations:

1. Fig 1, subpanel f: it would help readers if the red and green dashed lines were labeled on the figure rather than in the caption. Red and green are difficult to distinguish for colorblind readers, but a small label next to each line would help.
2. Figure 2(a,b): The most noticeable feature in the temperature dependence of the photon energy is a big kink near 250K in CrSBr and at 200K in CrSBr/FGT. To my untrained eye, this is more noticeable than the kink at 130K. What is it? This needs to be commented upon, but it is not discussed.
3. Line 345: What does "at the momentum vector" mean? Which momentum vector? The vector where phonons and magnons hybridize? If so, state that explicitly.
4. Conclusion: "The photon energy of CrSBr exhibits selective tunability". This seems rather an exaggeration, as Fig. 2 shows that temperature shifts the exciton energy much more than layering. Indeed, the effect of layering is fairly minor, on the single meV level, whereas temperature has a 50 meV effect. Just so casual readers don't get the wrong idea, please explicitly state the percent "tunability" that layering produces in the CrSBr exciton. I am also quite skeptical of some of the other claims that e.g. this will be useful for dynamic tuning of laser output wavelengths. Please tone down the conclusion to realistic claims, or else back up those that seem fantastical.

In conclusion, I find the experimental results valid and worth publishing, but the discussion of applications seems decoupled from the actual magnitude of the effects measured. Perhaps I am ignorant of these applications, and a 6 meV shift at 100 K is really something new which will enable a technological breakthrough in photospintronics. But the reader needs more convincing. Whether this constitutes a significant enough advance for Nature Communications, I leave as a decision for the editor.

Version 1:

Reviewer comments:

Reviewer #2

(Remarks to the Author)

The authors have addressed all my concerns. I recommend it for publication.

Reviewer #3

(Remarks to the Author)

Lan et al have resubmitted their manuscript on exciton coupling to magnons in a CrSBr/Fe₃GaTe₂ heterostructure. They have sufficiently addressed all my technical comments, but I remain concerned about the overall impact of their results and whether their claims are overstated given the very small effect they report. They still have hyperbolic language in the text: "a breakthrough observation", etc. which seems to me unjustified. They have failed to explain why a 6 meV shift in a 1.3 eV feature (a 0.5% effect) that is 100 meV wide is enough to constitute a breakthrough or be practically useful for spintronics. I can imagine that if this effect could be enhanced this would be useful--but as it is, these results on their own seem neither particularly surprising or impressive.

On that note, I must insist on my previous request: that the authors explicitly state in the abstract and conclusion the percent modulation of the exciton energies they observe relative to the total bandwidth. The authors stated instead the percent of the shift of magnetic field, which is not what I asked. The casual reader who glances through just the abstract and conclusion needs to know exactly how large this effect is. Transparency about the magnitude is of utmost importance.

These results warrant publication somewhere, but unless the authors can clearly explain why their effect is as useful and breakthrough as they claim, it is perhaps not appropriate for the wide readership of Nature Communications.

Version 2:

Reviewer comments:

Reviewer #3

(Remarks to the Author)

I thank the authors for putting the shift relative to total bandwidth in the abstract. I now am satisfied that my previous concerns have been met.

Reviewer #4

(Remarks to the Author)

The manuscript entitled "Spin hardening-boosted coherent magnon-exciton coupling in van der Waals heterostructure" represents an impressively wide collection of experimental results comparing the properties of CrSBr and CrSBr/FGT heterostructures. The experiments are of high quality. However, I agree with previous referees that the broad implications of the results are overstated, and I find that the totality of the results do not seem to come together into a coherent story.

The first main deficiency of the paper is that it asserts a lot of mechanisms for the observed properties without distinguishing those mechanisms from other possibilities or demonstrating consistency with the experiments through e.g. the use of simulations or models. Unless it can be made precise what their mechanisms are predicting, it is not clear that they are correct. The second main deficiency is that many terms are ill-defined (e.g. spin-hardening, tripartite interaction), and ultimately come across more as a phenomenological description of the experimental data rather than a clearly defined microscopic explanation. Sometimes different parts of the manuscript seem to be referring to different mechanisms for the same observation, which further adds to the confusion. The third deficiency is the overall results are underwhelming: the main conclusion that a small shift of the exciton PL, increase in saturation field, etc. can be induced by interfacial effects is not surprising.

With these comments, I do not recommend publication in Nature Communications.

To aid the discussion I provide a detailed discussion of some aspects, which align with some comments of previous referees.

1. What is the meaning of "phonon-mediated magnon-exciton tripartite interactions"?

It seems the temperature dependence of the PL and Raman data in Fig. 2 can be attributed to regular thermal expansion and magneto-elastic coupling. It is clear that the exciton energy may be influenced by the local crystalline environment; Cr(III) complexes display a wide range of PL emission energies ... Cr(III) PL in rubies is of course famously used for in-situ pressure sensors. It is clear the structure has a temperature dependence, through anharmonicity and in response to the magnetic correlations. These effects would be expected to be modified in the heterostructures due to differences in thermal expansivity and magneto-elastic coupling in combination with interfacial strain or charge transfer, so that CrSBr and CrSBr/FGT would be expected to display slightly different temperature dependence of the PL and phonon Raman response. I do not find the observations surprising.

Is the author's scenario the same or different from these conventional effects?

If it is the same, there is no need for a new name for these effects.

If it is different, the authors should indicate how it differs from conventional effects, present theory showing how it may be distinguished, and demonstrate it experimentally.

2. Is the "spin hardening" a consequence of exchange coupling between the FGT and the CrSBr, or a modification of the magnetic anisotropy (for example via charge transfer or strain)?

First, it should be noted that the magnetic anisotropy of FGT and related compounds is sensitive to both the Fe valency and strain (see discussion in Wu, S. et al Nature Communications 15, 10765 (2024)).

Second, the increase of the saturation field of CrSBr when in contact with FGT seems to scale linearly with the magnetic order parameter of CrSBr, which would be the expected result if the magnetic anisotropy of CrSBr were increased or exchange between CrSBr layers was increased. If it were the direct result of exchange coupling with the FGT, it would be expected to scale with the magnetization of the FGT, which it does not.

The discussion of this effect is confusing in the text.

For example, page 12 reads: "the interfacial charge transfer can simultaneously modify the electronic structure of Cr, thereby exerting a profound impact on macroscopic magnetic properties. Specifically, the increased electron occupancy in Cr-3d orbitals of CrSBr enhances their electron cloud density and amplifies the spatial overlap of adjacent Cr atomic orbitals[52-54]. As a result, the strengthened orbital hybridization and Cr-Cr AFM coupling further stabilize the AFM spins along the zigzag chains and enhance the magnetic proximity interactions between CrSBr and FGT, collectively contributing to the spin hardening and the suppressed lattice vibration."

Here it seems to imply that charge transfer results in a modification of the magnetic couplings within the CrSBr.

However, the conclusion reads "We demonstrate that the robust interfacial magnetic exchange interactions from FGT harden the AFM spin ordering in CrSBr, which therefore modifies the vibrational polarity of magnetic ions and elevates the spin-flop field."

Here it seems to imply magnetic exchange interactions between FGT and CrSBr, not charge transfer. What do the authors mean? What is meant by vibrational polarity?

3. I do not view the collected data as unique indication of coherent exciton-magnon coupling.

First, demonstrating that the PL is influenced by the magnetic state of the material is not sufficient to imply coherent exciton-magnon coupling; as noted above, the PL may be influenced by effects like strain and charge transfer, which are clearly implied by the other measurements. The fact that the structure (and therefore every other property) is sensitive to the magnetic ground state through magneto-elastic coupling is not the same as coherent coupling of the exciton to magnetic excitations above the ground state. Here is a place where an explicit model, with precise predictions for the temperature or field dependence, should have been employed to justify the uniqueness of the conclusions of the manuscript.

Second, the PL in CrSBr has previously been demonstrated to have significant vibrational fine structure (see Lin et al, ACS Nano 2024, 18, 2898, which should have been cited in the present work). Much of the observed lineshape is likely dominated by this fine structure. It is more accurate to say that the lineshape (including the large vibrational contribution) has been modified by FGT, but it is not clear the fundamental exciton has shifted. In Fig. 2c there are clearly differences in the lineshapes and linewidths, making the peak position itself a complicated measure of any effect.

Third, it must be remarked that the slope of the PL peak energy as a function of temperature remains similar for $T > T_N$ and $T < T_N$. However, for $T > T_N$ there are no coherent magnons to couple with. It is stated in the main text: "we perform a fitting analysis of the exciton peak shift using the power-law dependence of the AFM transition and yield a well-fitted result (Fig. S5), which further provides compelling evidence for coherent magnon-exciton coupling in the heterostructure." However, the power law fitting of the peak position in Fig. S5 is not very compelling at all. The data shows a continuous shift of the peak energy that is considerably more broad than the fitted function, suggesting there are mechanisms other than magnetic ordering contributing to the shift.

As an additional comment: the extraction of the peak energy itself does not appear consistent across different figures. What is the meaning of the black lines in Fig. 2(a,b)? Why do they not match the peak positions plotted in Fig. 2d? Why do the positions of the arrows in Fig. 2c not correspond to the maxima of the curves, and also not match the peak positions in Fig. 2d? (e.g. the arrows at 250 and 200K in Fig. 2c appear to be at the same energy, but they are not in Fig. 2d).

Also: the caption of Fig. S5 says that it was fit to a function with $I^{\wedge}(2w)$, but this is not second harmonic generation; it is PL peak position. It is not clear why $I^{\wedge}(2w)$ appears in this expression. The same expression appears in the caption of Fig. S12, which addresses Raman phonon peak intensity, which is also not $I^{\wedge}(2w)$.

4. I do not see the emphasized finding of bidirectional modulation of PL peak energy to be very remarkable.

Cr(III) compounds show a range of PL energies. It can be expected that the PL peak energy can be modified by chemistry, strain, charge doping, etc. The stated novelty of some recent works in this area has been the observation that external fields may additionally be used to control the PL peak energy in a single sample at a fixed temperature. The present finding that the PL peak can also be tuned by interfacial effects is not in the same category as these recent works, because it cannot be controlled by external stimuli. The shifts are also a small percentage of the bandwidth, which limits the utility, as previous referees have noted.

Version 3:

Reviewer comments:

Reviewer #4

(Remarks to the Author)

The authors have made a sincere and good faith effort to respond to the comments of all referees, and have added a significant amount of new analysis. The clarified descriptions make it possible to address the accuracy of their interpretation with follow-up studies.

As Reviewer #4, I was asked to provide a first review after several rounds of review had occurred, and most of the points I raised were intended to confirm the validity of the same concerns raised by the other referees in previous rounds. Those other referees have now indicated they are satisfied. While the authors have not fully assuaged all of my concerns regarding the interpretation of the collected data, it is not fair to the authors to relitigate those concerns.

On this basis, I recommend to accept.

Response to Reviewers and a Summary of Changes

Many thanks to the reviewers for having given us valuable comments on the manuscript (NCOMMS-25-47700-T) submitted to *Nature Communications*.

Title: Spin hardening-boosted coherent magnon-exciton coupling via phonon-mediated tripartite interactions in van der Waals heterostructure

Authors: Weican Lan, Chaocheng Liu, Ruiqi Liu, Yafei Chu, Lu Cheng, Yajuan Feng, Chao Wang, Huijuan Wang, Minghui Fan, Yuran Niu, Hengli Duan, Wensheng Yan

We deeply appreciate the valuable feedback and comments provided by the reviewers, as well as being grateful for their positive comments on this work: “I find the results rather interesting and all experimental data appears to be of high quality” (Reviewer #1); “These findings hold significant importance for exciton physics and spintronics”, “I believe this work is interesting and suitable for publication in Nature Communications” (Reviewer #2); “Their experiments are of high quality and contain careful analysis”, “I find the experimental results valid and worth publishing” (Reviewer #3).

The reviewers presented professional comments and revision suggestions before the paper could be reconsidered for publication in *Nature Communications*. Their insights have been invaluable in guiding us towards enhancing the quality and clarity of our manuscript. We have seriously considered these suggestions and comments, made the revision accordingly, and would like to resubmit the revised manuscript. We sincerely appreciate the hard work of the reviewers and hope that the revised manuscript can meet the acceptance criteria. The detailed responses to the comments are presented in a point-to-point manner as follows. Accordingly, all responses to reviewers' comments are highlighted in blue; all revisions in the manuscript and supplementary files are highlighted in red.

Response to Reviewer #1

The authors report measurements of the PL spectrum of CrSBr/Fe₃GaTe₂ heterostructures. They observe a strong dependence of the temperature and a steep rise of the main PL peak when moving below the Neel temperature of CrSBr in CrSBr/FGT heterostructures. The results are interpreted in terms of coupling to magnons and phonons. **I find the results rather interesting and all experimental data appears to be of high quality.** However, I am far from convinced on the general interpretation of the results and cannot recommend the manuscript for publication in Nat. Commun.

We sincerely appreciate the reviewer's careful attention to our results and the recognition of our experimental data quality. We fully understand the reviewer's concerns about our overall interpretation. In response, we have carefully revised the manuscript, supplemented relevant evidence, and performed more in-depth analyses to support the coupling between magnons, excitons, and phonons. We believe these revisions have significantly enhanced the clarity and reliability of our conclusions. We hope the revised manuscript now meets the high standards of *Nature Communications*, and thank the reviewer once again for your valuable feedback.

Comment: The authors mention several times that charge transfer occurs between the FGT and CrSBr. Why is that not enough to explain the spin hardening and change in PL spectrum? There is a clear (but) weak coupling between the exciton energies and the magnetic order. But to refer to this as exciton magnon coupling is stretching it in my opinion. The excitons live at completely different energy scales compared to the magnons. Demonstrating dependence of PL on magnetic order is not the same as demonstrating exciton-magnon coupling. Along the same lines: the spin hardening could again be described by charge transfer right? So I cannot see that it is signalling interlayer magnetic interactions. Finally, the phonons are observed to depend on the magnetic order as well. In this case it is perhaps expected to have some degree of magnon-phonon coupling, but I cannot see any explicit evidence of it.

Response: Thanks for your professional and valuable comments. In our constructed CrSBr/FGT heterostructure (semiconductor CrSBr/metal FGT), charge transfer persists across the entire temperature range. However, a significant PL peak shift in CrSBr is only observed below its T_N (the magnetic ordering temperature of CrSBr). When the temperature exceeds T_N , the CrSBr PL exhibits the same temperature-dependent behavior as pristine CrSBr, without any abrupt transition in PL peaks. Therefore, it is clearly evident that charge-transfer-driven effects are insufficient to explain the observed spin hardening and exciton changes in CrSBr. Instead, the PL peak shift in

CrSBr exhibits a close correlation with magnetic order under the effect of ferromagnetic substrate FGT. To further validate this conclusion, we conducted a comparative experiment. We fabricated a heterostructure by stacking the **nonmagnetic metallic** material WTe₂ (whose conductivity is comparable to that of FGT [*Nature* 514, 205-208 (2014); *Chin. Phys. Lett.* 40, 087501 (2023)]) with CrSBr. The KPFM results (Fig. R1a) demonstrated that charge transfer still occurs between the two materials in the same direction. However, in the absence of interfacial magnetic interactions, the PL spectra exhibit no correlation with the magnetic order of CrSBr (Figs. R1b and R1c). This further confirms that the exciton changes in CrSBr cannot be fully explained by charge transfer alone, but are primarily attributed to its magnon-exciton coupling modulated by FM FGT.

The intimate correlation manifested between excitons and magnons is generally referred to as the coupling effect. Magnons, the quantized excitations of collective spin waves, appear only in systems with long-range spin order (i.e., below the magnetic transition temperature). Notably, this spin-order-dependent magnon–exciton interaction is macroscopically manifested as a significant dependence of excitonic energy on magnetic order, which has been widely reported in numerous pioneering works, encompassing both single systems and heterostructures [*Nature* 609, 282-286 (2022); *Nat. Mater.* 20, 1657-1662 (2021); *Nature* 583, 785-789 (2020); *Nano Lett.* 20, 4625-4630 (2020); *Adv. Mater.* 35, 2300247 (2023)]. In this context, excitons and magnons possess intrinsic coherence, termed magnon-exciton coupling. We certainly agree with your viewpoint that excitons and magnons typically occupy distinct energy regimes. Notably, Xiaodong Xu and Xiaoyang Zhu et al. explicitly state in their *Nature* article: "*Although magnons and excitons are energetically mismatched by orders of magnitude, their coupling can lead to efficient optical access to spin information*" [*Nature* 609, 282-286 (2022)], and they concurrently elucidate coherent magnon-exciton coupling behavior in CrSBr. Undoubtedly, magnons and excitons can undergo coupling interactions, leading to spin order/magnon-dependent excitonic energy shift. Such magnon-exciton coupling behavior, manifesting as the dependence of PL on magnetic order, has been extensively verified in CrSBr [*Nature* 620, 533-537 (2023); *Nat. Mater.* 20, 1657-1662 (2021); *Nat. Nanotechnol.* 18, 23-28 (2023); *Nat. Mater.* 24, 391-398 (2025)]. On the other hand, regarding the role of charge transfer you are concerned about, we do not completely rule out its influence on the PL peak shift. The charge transfer occurring in CrSBr/FGT heterostructure cannot adequately account for this magnon-exciton coupling behavior, yet it does play a pivotal role in mediating the anomalous excitonic PL peak shift. The charge transfer promotes interfacial magnetic interactions, leading to spin hardening and phonon vibration confinement. Consequently, as we discussed in the main text, this process breaks the magnon-exciton coupling constraints in CrSBr under the first-order perturbation theory (FPT),

ultimately inducing a blueshift in exciton energies. Therefore, we firmly believe that the observed spin-order-dependent PL peak shift in the CrSBr/FGT heterostructure is predominantly governed by magnon-exciton coupling, with charge transfer providing synergistic enhancement. More critically, we further perform a fitting analysis of the exciton peak shift in CrSBr using the power-law dependence of the AFM transition and yield a well-fitted result (Fig. R2), which further provides compelling evidence for coherent magnon-exciton coupling in the heterostructure.

Similarly, we also acknowledge the fact that charge transfer contributes to spin hardening. We have claimed in the main text that orbital-mediated charge transfer assists in not only orientationally enhancing the spin ordering (spin hardening) in CrSBr but also reciprocally amplifying the magnetization of FGT. However, despite the presence of charge transfer effects, interfacial magnetic interactions remain the dominant mechanism governing spin hardening. Specifically, the increased electron occupancy in Cr-3d orbitals of CrSBr enhances their electron cloud density and amplifies the spatial overlap of adjacent Cr atomic orbitals [*J. Am. Chem. Soc.* 146, 34070-34079 (2024); *ACS Nano* 17, 22684-22690 (2023); *Nat. Commun.* 16, 3691 (2025)]. As a result, the strengthened orbital hybridization and Cr-Cr AFM coupling further stabilize the AFM spins along the zigzag chains and enhance the magnetic proximity interactions between CrSBr and FGT, collectively contributing to the spin hardening. Consequently, we conclude that the spin hardening in CrSBr results from interfacial magnetic interactions, with a contribution from charge transfer. Actually, we have presented sufficient analysis and results to demonstrate the interfacial magnetic interactions in CrSBr/FGT heterostructure. For instance: the dependence of Raman modes and PL peaks on magnetic order (Fig. 2); the enhanced spin-flop fields and coercive fields observed in magnetotransport and Hall measurements (Fig. 3); and the characterization using synchrotron-based XAS and XPEEM (Fig. 4). Therefore, the interfacial magnetic interactions definitely exist in our CrSBr/FGT heterostructure, evidencing their universal presence across all van der Waals magnetic heterostructures [*Nat. Mater.* 22, 305-310 (2023); *Nat. Mater.* 23, 212-218 (2024); *Nat. Commun.* 11, 6021 (2020); *Nat. Commun.* 14, 2190 (2023); *Nat. Nanotechnol.* 16, 856-868 (2021); *Adv. Mater.* 37, 2413438 (2024); *Nat. Mater.* 22, 1311-1316 (2023)].

As you mentioned, the Raman modes of CrSBr also exhibit a close dependence on magnetic order. In the CrSBr/FGT heterostructure, the A_{2g} mode of CrSBr is strongly suppressed below its T_N . However, when CrSBr enters a paramagnetic state above T_N , the A_{2g} mode approximately aligns with that of pristine CrSBr. This result directly demonstrates that magnetic exchange interaction between AFM CrSBr and FM FGT only occurs below T_N . Consequently, due to the interfacial magnetic interaction between the FGT and CrSBr, the Raman mode changes in CrSBr present a magnetic-order dependence. This signature feature is widely recognized as spin-phonon coupling,

as generally reported in numerous pioneering studies, including in the CrSBr system [*Sci. Adv.* 7, eabj3106 (2021); *Nat. Commun.* 10, 345 (2019); *Phys. Rev. B* 107, 075421 (2023)]. Furthermore, to more intuitively visualize the magnetic-order-dependent phonon mode variation, we further perform a fitting analysis of the A_{2g} mode in CrSBr within the heterostructure. The acquired results demonstrate that its evolution follows AFM transition power-law behavior (Fig. R3), providing compelling explicit evidence for spin-phonon coupling in our CrSBr/FGT heterostructure.

Overall, we have ruled out the charge transfer as the sole factor governing spin hardening and PL spectral changes, establishing robust evidence for interfacial magnetic interactions in the CrSBr/FGT heterostructure. We have demonstrated the existence of both magnon-exciton coupling and spin-phonon coupling, and identified that phonon-mediated magnon-exciton coupling in CrSBr is the fundamental origin of the anomalous PL blueshift. Once again, we sincerely appreciate your thorough review and constructive comments, which greatly contributed to the improvement of our paper.

Fig. R1. KPFM and temperature-dependent PL for CrSBr/WTe₂. **a**, KPFM topography and work function plots of CrSBr/WTe₂ heterostructures. The blue and purple dashed lines mark the areas of CrSBr and WTe₂, respectively. Scale bar, 5 μm. **b**, Temperature-dependent PL spectra for pristine CrSBr and CrSBr/WTe₂. The triangles correspond to the energy position of their PL peaks. **c**, Variations of the PL peak positions for pristine CrSBr and CrSBr/WTe₂ under different temperatures.

Fig. R2. Magnetic-order-dependent exciton fitting. Variations of the PL peak positions for CrSBr/FGT under different temperatures. The red solid curve is a best fit to the power law function $I^{2\omega} \propto [a + b(T - T_N)\beta]^2$.

Fig. R3. Magnetic-order-dependent phonon fitting. The intensity variation of A_g^2 modes for CrSBr/FGT under different temperatures. The red solid curve is a best fit to the power law function $I^{2\omega} \propto [a + b(T - T_N)\beta]^2$.

Revision:

Accordingly, Figs. R1-R3 with relevant captions have been added on pages 6, 8, 12 of the Supplementary Materials.

The supplementary description “More critically, we perform a fitting analysis of exciton peak shift using the power-law^[45] dependence of the AFM transition and yield a well-fitted result (Fig. S5), which further provides compelling evidence for coherent magnon-exciton coupling in the heterostructure.”; “To clarify the specific role of charge transfer in this process,”; “Furthermore, we fabricated a heterostructure by stacking the nonmagnetic metallic material WTe₂ (whose conductivity is comparable to that of FGT^[46,47]) with CrSBr. The KPFM results (Fig. S7a) demonstrated that charge transfer still occurs between the two materials in the same direction. However, in the absence of interfacial magnetic interactions, the PL spectra (Figs. S7b and S7c) exhibit

no correlation with the magnetic order of CrSBr. This further confirms that the exciton changes in CrSBr cannot be fully explained by charge transfer alone, but are primarily attributed to magnon-exciton coupling.” **have been added** at different positions on page 6 of the Main Text.

The supplementary description “We further perform a fitting analysis of the A_g^2 mode in CrSBr within the heterostructure. The acquired results demonstrate that its evolution follows AFM transition power-law behavior (**Fig. S11**), providing compelling explicit evidence for spin-phonon coupling in our CrSBr/FGT heterostructure.” **has been added** on page 7 of the Main Text.

To summarize, I find the results of the manuscript interesting, but not the interpretation and title. The title is difficult to understand for one thing and I do not find any evidence of its claim. What does coherent magnon-exciton coupling mean? Why do you believe this coupling is mediated by phonons? And why do you believe it is boosted by spin-hardening?

Response: We sincerely appreciate your efforts and dedication to our article, as well as being grateful for your comments on this work. We deeply apologize for any difficulties or confusion that our current manuscript may have caused. To address this, we have provided further clarification regarding our experimental findings and the discussion of our results.

In our work, we demonstrate a reciprocal interfacial magnetic interaction within the CrSBr/FGT heterostructure, where the excitonic PL peaks of CrSBr exhibit a close dependence on magnetic order. In particular, we observe an anomalous blueshift in the PL peaks, which we verify to arise from phonon-mediated coherent magnon-exciton coupling. Here, "coherent magnon-exciton coupling" refers to a specific quantum phenomenon where magnons (collective excitations of electron spins in magnetic materials) and excitons (bound electron-hole pairs in semiconductors) interact in a synchronized, energy-exchanging manner [*Nature* 583, 785-789 (2020); *Nature* 609, 282-286 (2022); *Nat. Nanotechnol.* 18, 23-28 (2023); *Nat. Phys.* 20, 801-806 (2024)]. Coherent coupling is a strong, resonant interaction where energy oscillates back and forth between magnons and excitons. It requires matching frequency scales and spatial overlap, and results in hybrid states with new quantum properties [*Nat. Phys.* 17, 1137-1143 (2021); *Phys. Rev. Lett.* 125, 267201 (2020); *Science* 379, 278-283 (2023)]. Notably, the characteristic dependence of exciton PL on magnetic order is commonly termed coherent magnon-exciton coupling, as widely investigated in numerous seminal works.

Regarding the involvement of phonons in such magnon-exciton coupling, we have also conducted an in-depth and meticulous analysis. Based on the magnetic-order-dependent changes in phonon modes, we identified the existence of spin-phonon

coupling in the CrSBr/FGT heterostructure. Moreover, this coupling effect constrains the phonon vibrations of magnetic atoms in CrSBr. As established, phonons can enhance magnon transport and propagation via magnon-phonon hybridization [*Sci. Adv.* 7, eabj3106 (2017); *Proc. Natl. Acad. Sci. USA* 112, 8977-8981 (2015)]. We therefore conclude that phonons can participate in the interaction between magnons and excitons. According to conventional FPT constrain, excitonic PL shift typically occurs only towards lower energies. However, in our system, apart from the restricted phonon vibrations in CrSBr, there are no other potential factors that could account for the anomalous blueshift of excitonic PL. Consequently, based on the comprehensive discussion and analysis above, the anomalous PL blueshift induced by magnon-exciton coupling can be primarily attributed to the phonon-mediated influence, which is theoretically/experimentally supported and permissible [*Phys. Rev. B* 108, 144425 (2023); *J. Phys. D: Appl. Phys.* 51, 224008 (2018); *Nano Lett.* 23, 9235-9242 (2023); *Nat. Commun.* 14, 1047 (2023); *Nat. Commun.* 14, 88 (2023)]. Moreover, we also perform the AFM transition correlated power-law fitting for CrSBr phonons (Fig. R3) and identify the spin-phonon coupling in the heterostructure, further highlighting the mediating influence of phonons on the magnon-exciton coupling. Therefore, we have compelling grounds to conclude that phonons are engaged in the magnon-exciton coupling, ultimately giving rise to the anomalous blueshift of PL peaks.

Through systematic characterization and analysis using magnetic, electrical, and spectroscopic techniques (Figs. 3 and 4), we confirm that the spins of CrSBr and FGT are both orientationally enhanced (spin-hardening) with significantly elevated spin-flop fields and coercive fields. We further unveil a positive correlation between spin-hardening and PL peak shift (see supplementary Fig. S15), confirming that higher spin hardening leads to a larger PL peak shift. This thus signifies that enhanced spin hardening will induce stronger magnon-exciton coupling, i.e., spin hardening-boosted magnon-exciton coupling. More precisely, the complete physical picture of magnon-exciton-phonon tripartite interactions in CrSBr is that charge transfer and interfacial magnetic interactions lead to spin-hardening and constrained phonon vibrations. These hardened spins simultaneously undergo magnon-exciton coupling while being influenced by vibrationally restricted phonons (via spin-phonon coupling), thereby resulting in an anomalous energy shift of the excitons.

In summary, based on the magnetic-order-dependent characteristic changes in Raman modes and excitonic PL, we confirm the presence of both spin-phonon coupling and magnon-exciton coupling in CrSBr, which are modulated by FM FGT. These conclusions are further substantiated by the power-law fitting of the AFM transition. By revealing and analyzing the positive correlation between spin-hardening and spin-flop, we establish that hardened spins promote magnon-exciton coupling in CrSBr. Moreover, through systematic analysis, we further confirm that phonons mediate this

magnon-exciton coupling process, resulting in the emergence of an anomalous PL peak blueshift in CrSBr. Once again, we are grateful for your professional comments and enthusiastic dedication, which have greatly helped us improve our manuscript.

Response to Reviewer #2

Excitons are of both fundamental interest and technological importance, showing great potential for applications in photospintronic and quantum technologies. In this manuscript, the authors investigate coherent exciton manipulation in a 2D CrSBr/Fe₃GaTe₂ heterostructure by utilizing phonon-mediated magnon-exciton interactions. Through a series of comparative experiments, including temperature-dependent Raman/PL, Hall effect, and XPEEM techniques, they demonstrate that interfacial magnetic coupling and orbital-driven charge transfer induce reciprocal spin hardening. This further leads to phonon-constrained magnon dynamics and a corresponding anomalous increase in exciton energy. These findings hold significant importance for exciton physics and spintronics, which provide a promising approach for modulating excitonic properties in 2D AFM semiconductors and can advance the development of spin-based quantum photonic devices. **I believe this work is interesting and suitable for publication in Nature Communications** after addressing several concerns:

Thank you for your thoughtful and constructive feedback on our manuscript. As you rightly highlighted, excitons can not only serve as a rich platform for fundamental research but also hold great potential for applications in opto-spintronics and quantum technologies. In this study, we designed a 2D CrSBr/Fe₃GaTe₂ heterostructure to investigate the coherent exciton modulation via phonon-mediated magnon-exciton interactions. We believe our findings offer important insights into exciton physics, quantum photonic devices, and spintronics in 2D systems, providing a novel approach to modulate excitonic behavior in vdW AFM semiconductors. We are grateful for your recognition of the innovation and significance of our work, and we will carefully address all remaining concerns to ensure the manuscript meets the high standards of *Nature Communications*. Below are our detailed responses to your comments.

Comment 1) In transport measurements, the FGT/CrSBr heterostructure is regarded as an electrically integrated system. However, it remains unclear whether the observed Hall response exclusively originates from the FGT layer. Given the pivotal role of interface effects in this study, it is important to deconvolute other potential contributions to the Hall signal in the heterostructure. Therefore, the possible influence of CrSBr may be considered. Clarifying this point is critical for a more accurate understanding of the magnetotransport mechanism in the system.

Response: We thank the reviewer for this insightful comment. To exclude the possible contribution from CrSBr to the Hall response in the heterostructure, we performed

control Hall measurements on devices consisting solely of CrSBr flakes, using the same electrode configuration and measurement conditions (Fig. R4). The results show that CrSBr alone exhibits no measurable Hall signal within the noise level of our measurement system, which is consistent with its AFM semiconducting nature and its negligible carrier density at low temperatures [Adv. Mater. 32, 2003240 (2020)].

These results confirm that the Hall response observed in the FGT/CrSBr heterostructure originates exclusively from the FGT layer. Therefore, the changes in the Hall signal can be attributed to the interfacial magnetic coupling and spin modulation induced by CrSBr, rather than to any direct transport contribution from CrSBr itself.

Fig. R4. Hall measurement of CrSBr. R_{xy} of the pristine CrSBr versus magnetic field at 5 K, measured with a perpendicular applied magnetic field. The inset shows an optical image of the CrSBr device. Scale bar, 20 μm .

Revision:

Accordingly, Fig. R4 with the relevant caption has been added on page 18 of the Supplementary Materials.

The supplementary description “To clarify the signal origin, we independently measured the Hall response of CrSBr and found a negligible Hall resistance (Fig. S17), demonstrating that the Hall signals observed in the heterostructure originate solely from the FGT layer.” has been added on page 9 of the Main Text.

Comment 2) Generally, within conventional FM/AFM heterostructures, exchange bias phenomena are frequently observed due to interfacial spin coupling. It would be informative for the authors to deliberate on whether any such effects exist in the CrSBr/FGT system. This consideration provides a valuable insight into the nature of the FM/AFM interface in this heterostructure.

Response: We sincerely appreciate the reviewer for raising this insightful question regarding the possible emergence of exchange bias in the CrSBr/FGT heterostructure. Exchange bias is a well-known interfacial phenomenon that typically arises in conventional ferromagnet/antiferromagnet (FM/AFM) heterostructures. It manifests as

a horizontal shift in the magnetic hysteresis loop and is commonly attributed to unidirectional anisotropy induced by interfacial exchange coupling between pinned AFM spins and switchable FM moments [*Phys. Rev.* 105, 904 (1957)]. The establishment of exchange bias generally requires two key conditions: (1) strong interfacial coupling between the FM and AFM layers, and (2) aligned or at least non-orthogonal magnetic anisotropies between the two components, which facilitates coherent spin pinning across the interface [*Nat. Commun.* 14, 2190 (2023); *Nano Lett.* 20, 5030 (2020); *Adv. Mater.* 32, 2002032 (2020)].

In our CrSBr/FGT heterostructure, although strong interfacial magnetic interactions are indeed present, evidenced by the observed spin hardening and phonon-constrained magnon dynamics, the magnetic easy axes of the two constituents are nearly orthogonal. FGT exhibits out-of-plane magnetization, while CrSBr possesses a strong in-plane antiferromagnetic anisotropy (as shown in Figs. 1b, 1c of the manuscript). This orthogonal spin configuration is generally unfavorable for the formation of a stable unidirectional exchange field, thereby suppressing the conventional exchange bias effect.

Consistent with this interpretation, our magnetization measurements (as shown in Figs. 3i, 3j of the manuscript) show no significant loop shift or coercivity asymmetry. Nevertheless, we believe the distinct spin configuration at the CrSBr/FGT interface offers an alternative pathway for novel interfacial spin phenomena, such as spin hardening and coherent magnon-exciton coupling, which are not typically accessible in conventional FM/AFM systems.

Comment 3) In Figure 3j, the author should explain why the vertical-axis values of the heterostructure do not align with those of the FGT reference in Figure 3i. Moreover, the magnitude of the anomalous Hall resistance in the heterostructure is significantly reduced compared to that of the pristine FGT layer, which also needs a mechanistic explanation for this discrepancy. These clarifications are beneficial to strengthen the interpretation of interlayer interactions and their influence on spin-dependent transport properties.

Response: We appreciate the reviewer's careful observation and insightful comment. The difference in the vertical-axis values of the Hall resistance between the pristine FGT and the FGT/CrSBr heterostructure arises from the presence of the additional CrSBr layer. Since CrSBr is a semiconducting material with significantly higher resistivity compared to metallic FGT [*Adv. Mater.* 34, 2109759 (2022); *Nat. Commun.* 13, 5067 (2022)], and it increases the overall resistance of the heterostructure when incorporated into the vertical device stack. The reduced magnitude of the anomalous Hall resistance in the heterostructure can be attributed to the interlayer charge transfer from CrSBr to FGT. This charge transfer decreases the number of unpaired Fe 3d

electrons in the FGT layer (as shown in Fig. 4b of the manuscript), thereby reducing the net magnetic moment and consequently weakening the anomalous Hall response. This explains why the red curve (heterostructure) in Fig. 3l is consistently lower in magnitude than the purple curve (pristine FGT). Nevertheless, this reduction in absolute Hall resistance does not affect the characteristic changes associated with magnetic transitions, which are governed by the interfacial magnetic coupling between CrSBr and FGT.

Comment 4) It shows that although the long-range AFM order in CrSBr disappears around 130 K, certain magnetic and optical features persist beyond this temperature. Specifically, the enhanced coercive field in the heterostructure and the increased exciton energy splitting remain observable up to approximately 200 K. This apparent lagging behavior suggests the presence of underlying magnetic correlations or interfacial effects that endure above the Néel temperature. It is advisable for the authors to clarify the possible physical origins of these effects.

Response: We appreciate the reviewer's careful reading and valuable suggestions. To clarify, CrSBr is a typical layered A-type antiferromagnet with in-plane AFM ordering [*J. Magn. Magn. Mater.* 92, 129. (1990)]. Above its Néel temperature (~130 K), CrSBr does not immediately transition into a conventional paramagnetic state. Instead, it experiences a narrow temperature range within which the long-range AFM order vanishes, while some residual short-range magnetic correlations still persist to some degree. These residual short-range magnetic orders have been reported in previous studies [*Nat. Mater.* 21, 754 (2022); *Adv. Mater.* 32, 2003240 (2020); *Nano Lett.* 21, 3511 (2021)] and are understood as quasi-static magnetic features, which may cause certain observable magnetic and optical responses even beyond the Néel temperature.

In our CrSBr/FGT heterostructure, although the system exhibits overall spin hardening, the strong magnetic exchange coupling at the interface with ferromagnetic FGT may induce localized ferromagnetic ordering or enhance magnetic correlations within CrSBr. This interfacial effect likely explains the persistent observation of the enhanced exciton energy and increased coercive field up to approximately 200 K. This "lagging" magnetic behavior suggests that interface-driven local magnetic order or residual short-range correlations extend beyond CrSBr's intrinsic magnetic transition temperature, thereby maintaining certain magnetic and optical features above the Néel temperature.

Revision:

Accordingly, the supplementary description "Within these temperature ranges, CrSBr/FGT exhibits even lower photon energies compared to pristine CrSBr, achieving a maximum difference of around 200 K. This observation indicates that the excitonic

response in the heterostructure remains sensitive to interfacial magnetic effects over an expanded temperature range, far exceeding the T_N of CrSBr. This suggests that the magnetic proximity effect and residual short-range magnetic correlations^[25,26,30] induced by the FGT layer persist well above the intrinsic magnetic transition temperature of CrSBr.” **has been added** on page 6 of the Main Text.

Comment 5) In Fig. 4i and Fig. S18, the authors should give a detailed discussion on why the XMCD-PEEM images show no discernible signal, and the XMLD-PEEM images reveal a single domain. Additionally, more detailed configuration information should be included in the Methods section regarding how the XMCD and XMLD images were acquired and analyzed.

Response: Thanks for your professional comments. All XPEEM data were recorded on MAXPEEM beamline at MAX IV Lab, after cooling from room temperature to 100 K under zero magnetic field with normal incidence (90°) of X-ray on the sample surface. This setup is sensitive to out-of-plane FM and in-plane AFM but cannot detect the in-plane FM domain. Fe and Cr XMCD-PEEM raw images were recorded by 1 s exposure time with right (R) and left (L) circularly polarized light at Fe L_3 resonance at 707.3 eV and Cr L_3 resonance at 576.4 eV. The XMCD asymmetry is given by $(I^L - I^R)/(I^L + I^R)$. Cr XMLD-PEEM raw images were recorded by 5 s exposure time with linear horizontal (LH) and vertical (LV) polarized light at 575.4 eV, and the XMLD asymmetry is given by $(I^{LH} - I^{LV})/(I^{LH} + I^{LV})$. Therefore, the Cr XMCD-PEEM image in Fig. S22 shows no discernible contrast, which means there is no out-of-plane FM moment for Cr element in FGT/CrSBr heterostructure. The Cr XMLD-PEEM image in Fig. 4i shows dark contrast in CrSBr area, which means the b -axis orientation of the AFM moments in CrSBr. Meanwhile, we have switched the order of the original Fig. 4i and Fig. S22 to improve the overall logical sequence of the manuscript. All these revisions have been clearly highlighted in the revised manuscript.

Revision:

Accordingly, the relevant experimental parameters and conditions “All XPEEM data were recorded after cooling from room temperature to 100 K under zero magnetic field with normal incidence (90°) of X-ray on the sample surface. This setup is sensitive to out-of-plane FM and in-plane AFM but cannot detect the in-plane FM domain.” have been thoroughly described on page 11 of the Main Text.

The supplementary description “The XPEEM measurement was carried out at the PEEM endstation of the MAXPEEM Beamline at the MAX IV laboratory (Lund, Sweden). Fe and Cr XMCD-PEEM raw images were recorded by 1 s exposure time with right (R) and left (L) circularly polarized light at Fe L_3 resonance at 707.3 eV and Cr L_3 resonance at 576.4 eV. The XMCD asymmetry is given by $(I^L - I^R)/(I^L + I^R)$.

Cr XMLD-PEEM raw images were recorded by 5 s exposure time with linear horizontal (LH) and vertical (LV) polarized light at 575.4 eV, and the XMLD asymmetry is given by $(I^{LH} - I^{LV})/(I^{LH} + I^{LV})$ ” has been added on page 19 of the Methods section in the Main Text.

Comment 6) A minor question. It is suggested that error bars should be added in Fig. 3e, 3f, 3k, and 3l to more precisely present the evolution process of the magnetic and magnetotransport properties as the temperature varies.

Response: We thank the reviewer for this insightful suggestion. We agree that error bars would help to more clearly illustrate the evolution of magnetic and magnetotransport properties with temperature. Accordingly, we have added appropriate error bars in Figs. 3e, 3f, 3k, and 3l in the revised manuscript to reflect measurement uncertainties and improve the clarity of data presentation. The corresponding figure caption has also been updated on page 10 of the Main Text. We appreciate the reviewer’s attention to this detail.

Fig. 3. e-f, The extracted H_s (e) and MRR (f) vary with temperature. k-l, The extracted H_c (k) and R_{xy} (l) as a function of temperature.

Comment 7) While the paper is highly readable, there are still a few spelling issues. The abbreviation “XMCD-XPEEM” should be corrected to “XMCD-PEEM”. The term “flipping” should be more accurately expressed as “flopping” to more appropriately describe the gradual reorientation or tilting of spins under external fields or internal interactions.

Response: We sincerely thank the reviewer for their careful reading and constructive suggestions regarding spelling and terminology. In response to the reviewer’s comment, we have corrected the abbreviation “XMCD-XPEEM” to “XMCD-PEEM” throughout the manuscript. The corrected term “XMCD-PEEM” more accurately refers to X-ray magnetic circular dichroism measurements performed using photoemission electron microscopy, and reflects common usage in the field. Furthermore, we have revised the term “flipping” to “flopping” in the context of spin dynamics. We agree that “flopping” more accurately describes the gradual reorientation or canting of spins under external magnetic fields or interfacial exchange coupling, especially in systems with AFM or

canted spin structures. These refinements improve the scientific accuracy and clarity of our manuscript. In addition to the grammatical and typographical issues kindly pointed out by the reviewer, we have also carefully checked the manuscript and identified a few additional inaccuracies and inconsistencies on our own. All revisions are highlighted in red in the revised manuscript. Thank you once again for your kind review and helpful suggestions.

In summary, I consider that the paper is interesting and has important guiding implications for the fields of spintronics and optoelectronics, but some minor revisions should be made to clarify the mentioned ambiguities and to meet the quality standards of NC.

We sincerely appreciate your high recognition of our work and deeply appreciate your efforts and dedication. Based on your professional comments and suggestions, we have carefully revised our manuscript. We eagerly hope that the revised version can meet your requirements and satisfy the acceptance criteria of NC.

Response to Reviewer #3

Lan et al present an experimental paper on the excitons of CrSBr layered on Fe₃GaTe₂. They find evidence of phonon coupling to magnetic order, exciton energy shifts with layering, and many other modifications to the CrSBr excitonic behavior. While **their experiments are of high quality and contain careful analysis**, I have a concern that their claims of tunability are somewhat overstated. Reading the abstract and conclusion, one would expect that they have found a way to control exciton energy over a wide range of energies--when in reality the shift is quite modest.

We would like to express our sincere gratitude to you for your insightful comments and suggestions. Upon careful reconsideration, we recognize that certain statements in the abstract and conclusion may inadvertently overstate the extent of the observed energy shifts, potentially giving the impression of large-scale spectral modulation. Therefore, we have revised the manuscript to clearly specify the extent of exciton energy modulation and objectively analyze its promising potential. We clarify that although the tuning range is limited compared to external apparatus such as electric/magnetic fields, this approach offers a distinctive advantage: it enables fine control of exciton energies without requiring external fields. We believe this constitutes a novel and potentially practical strategy for excitonic modulation, which could be valuable for future device applications. Once again, we sincerely appreciate the reviewer's positive evaluation and constructive feedback.

Comment 1) Fig 1, subpanel f: it would help readers if the red and green dashed lines were labeled on the figure rather than in the caption. Red and green are difficult to distinguish for colorblind readers, but a small label next to each line would help.

Response: We thank the reviewer for the valuable suggestion regarding figure accessibility. In the revised version of Fig. 1f, we have added small text labels directly next to the red and green dashed lines to improve clarity for all readers, including those with color vision deficiency. We believe this adjustment enhances the readability and accessibility of the figure. The revised Fig. 1f has been updated on page 5 of the Main Text.

Fig. 1f. The optical micrograph and thickness line profile of FGT/CrSBr heterostructures on SiO₂/Si substrate. The red and green dashed lines sketch the areas of CrSBr and FGT, respectively. Scale bar, 10 μm.

Comment 2) Figure 2(a,b): The most noticeable feature in the temperature dependence of the photon energy is a big kink near 250K in CrSBr and at 200K in CrSBr/FGT. To my untrained eye, this is more noticeable than the kink at 130K. What is it? This needs to be commented upon, but it is not discussed.

Response: We thank the reviewer for raising this insightful question. Here we provide a separate discussion on the two anomalous temperature points mentioned by the reviewer:

(1) “a big kink near 250K in CrSBr”. Although a certain degree of steepness can indeed be observed near 250 K in Fig. 2a, we attribute this feature primarily to the relatively large temperature intervals used during the measurement. Above 200 K, as the CrSBr system enters a paramagnetic state, we increased the temperature step to 50 K for the PL measurements. This wider interval leads to a less continuous data appearance and makes the exciton energy shift seem more abrupt. Nevertheless, the overall trend remains essentially linear within the margin of error. As shown in Fig. 2d, the evolution across this temperature range is smooth, with no clear sign of a sudden transition.

(2) “200K in CrSBr/FGT”. Indeed, as shown in Fig. 2(b,d,e), a pronounced kink can be observed around 200 K in CrSBr/FGT heterostructure, with a magnitude even greater than the change near the Néel temperature (~130 K). We have carefully analyzed this issue and provide the following discussion:

CrSBr is a prototypical layered A-type antiferromagnet, characterized by ferromagnetic coupling within each layer and antiferromagnetic alignment between adjacent layers. Above its Néel temperature, the system does not immediately enter a fully paramagnetic state. Instead, it undergoes a narrow transitional regime (~10 K) in which long-range antiferromagnetic order is lost, while short-range interlayer ferromagnetic correlations may persist. Prior studies [*Nat. Mater.* 21, 754 (2022); *Adv. Mater.* 32, 2003240 (2020); *Nano Lett.* 21, 3511 (2021)] have reported such short-range magnetic ordering, which often manifests as quasi-static magnetic features that remain observable in both magnetic and optical responses, exhibiting a form of “magnetic

memory” or hysteresis.

In the CrSBr/FGT heterostructure, the strong intrinsic ferromagnetism of FGT may induce or stabilize local ferromagnetic alignment within the CrSBr layer via interfacial exchange coupling, thereby extending the influence of short-range magnetic correlations. This effect may account for the abrupt shift in exciton energy observed near 200 K—well above the intrinsic magnetic transition temperature of CrSBr. Such a pronounced deviation suggests that interfacial magnetic interactions can significantly influence excitonic states over an expanded temperature range. The excitonic kink near 200 K may signal a threshold in the interfacial magnetic modulation, leading to a detectable optical response. Indeed, the behavior above the Néel temperature is more intricate. While our earlier analysis primarily focused on coupling features at low temperatures, we acknowledge that the anomalous excitonic behavior above the magnetic transition temperature was not sufficiently addressed in the original manuscript. Corresponding discussions have now been added in the revised version.

We sincerely appreciate the reviewer’s insightful comment, which has drawn our attention to this important temperature regime and helped us improve the clarity and completeness of our analysis.

Revision:

Accordingly, the supplementary description “Within these temperature ranges, CrSBr/FGT exhibits even lower photon energies compared to pristine CrSBr, achieving a maximum difference of around 200 K. This observation indicates that the excitonic response in the heterostructure remains sensitive to interfacial magnetic effects over an expanded temperature range, far exceeding the T_N of CrSBr. This suggests that the magnetic proximity effect and residual short-range magnetic correlations^[25,26,30] induced by the FGT layer persist well above the intrinsic magnetic transition temperature of CrSBr.” **has been added** on page 6 of the Main Text.

Comment 3) Line 345: What does "at the momentum vector" mean? Which momentum vector? The vector where phonons and magnons hybridize? If so, state that explicitly.

Response: We sincerely thank the reviewer for this thoughtful and important question. Indeed, in our original manuscript, the phrase “at the momentum vector” does not refer to an arbitrary momentum point, but rather to the specific point or narrow range in momentum space where the intrinsic dispersion curves of phonons and magnons intersect or come very close to each other. At these points, the proximity of their dispersion relations leads to enhanced magneto-elastic coupling, which in turn induces strong hybridization between the two excitations—forming the so-called resonant magnetoelastic mode. This process effectively facilitates faster propagation of magnons in the low-momentum region [*Phys. Rev. Lett.* 115, 197201 (2015); *Phys. Rev. Lett.* 117,

207203 (2016)]. We acknowledge that the original wording was imprecise and could cause confusion. Therefore, we have revised the manuscript to clarify this point accordingly and have included relevant references for support.

Revision:

Accordingly, the original description on page **14** of the Main Text “Furthermore, the propagation of magnons at the momentum vector is related to the hybridization between magnons and acoustic phonons through magneto-elastic coupling^[53-54].” **has been changed to** “Furthermore, the enhanced propagation of magnons arises at specific momentum regions where the intrinsic dispersion relations of magnons and acoustic phonons intersect or come into close proximity, enabling strong hybridization through magnetoelastic coupling^[56-59].”

Comment 4) Conclusion: "The photon energy of CrSBr exhibits selective tunability". This seems rather an exaggeration, as Fig. 2 shows that temperature shifts the exciton energy much more than layering. Indeed, the effect of layering is fairly minor, on the single meV level, whereas temperature has a 50 meV effect. Just so casual readers don't get the wrong idea, please explicitly state the percent "tunability" that layering produces in the CrSBr exciton. I am also quite skeptical of some of the other claims that e.g. this will be useful for dynamic tuning of laser output wavelengths. Please tone down the conclusion to realistic claims, or else back up those that seem fantastical.

Response: Thanks for your professional and valuable comments. In response to your concern regarding the magnitude of exciton energy modulation, we provide the following clarification:

We fully acknowledge that the exciton energy modulation achieved in our current experiments is relatively limited—approximately 8 meV, which is smaller than the temperature-induced exciton energy variation of up to 50 meV. However, it is worth noting that such a substantial temperature-driven shift requires an extremely broad temperature range (from 80 K to 350 K), which is inefficient for practical exciton energy control. Moreover, the redshift of exciton energy induced by temperature thermal effects is a common phenomenon in most material systems and, strictly speaking, cannot be regarded as control or modulation. In contrast, the layer-stacked CrSBr/FGT strategy presented in this work enables exciton energy modulation within a narrow temperature window near the magnetic phase transition, without requiring large temperature variations. This strategy demonstrates more efficient and flexible control capabilities in exciton energy through magnon-exciton coupling. On the other hand, to explicitly state the percent "tunability", we note that external magnetic field devices can achieve a maximum exciton energy shift of approximately 20 meV [*Nature* 609, 282-286 (2022); *Nat. Mater.* 20, 1657-1662 (2021)]. In this context, the 8 meV

shift achieved in our work can reach 40% of the modulation amplitude achievable with external magnetic fields. This represents a considerable degree of modulation. More importantly, our strategy does not require any external magnetic field or other complex instrumentation, thereby offering new pathways and application potential for exciton manipulation and utilization.

To avoid overstated claims in the original manuscript, we have revised the relevant expressions in the manuscript. For example, phrases such as “selective tunability of the photon energy in CrSBr” have been clarified to more accurately reflect the modulation magnitude and underlying physical mechanisms. Similarly, forward-looking statements like “this study can be used for dynamic tuning of laser emission wavelength” have been either moderated or removed, with additional qualifiers and contextual explanations added to better reflect their realistic significance. We sincerely thank the reviewer for the constructive feedback, which has helped us improve the rigor and accuracy of our manuscript.

Revision:

Accordingly, the original description on page 2 of the Main Text “Our findings uncover an accessible avenue for the tunable modulation of exciton energy in 2D AFM semiconductors, thus potentially addressing the demand for diverse wavelength light sources in quantum information and optoelectronic technology such as communication and sensing.” **has been changed to** “Our findings demonstrate an approach for modest modulation of exciton energy in 2D AFM semiconductors, which may provide insights for future exploration of wavelength control in quantum information and optoelectronic technologies such as communication and sensing.”.

The original description on page 15 of the Main Text “Notably, here the photon energy of CrSBr exhibits selective tunability depending on magnon-exciton coupling, unveiling significant potential in achieving dynamic tuning of laser output wavelengths.” **has been changed to** “Notably, the exciton energy shift achieved in our work can reach 40% of the modulation amplitude achievable with external magnetic fields^[32,35], indicating a substantial level of modulation and offering a valuable reference for future studies on wavelength control in laser systems.”.

In conclusion, I find the experimental results valid and worth publishing, but the discussion of applications seems decoupled from the actual magnitude of the effects measured. Perhaps I am ignorant of these applications, and a 6 meV shift at 100 K is really something new which will enable a technological breakthrough in photospintronics. But the reader needs more convincing. Whether this constitutes a significant enough advance for Nature Communications, I leave as a decision for the editor.

Sincere thanks for your recognition of our work. We deeply appreciate your efforts and dedication to our article, as well as being grateful for your comments on this work. In accordance with your expert comments and suggestions, we have meticulously revised the manuscript. We sincerely hope this revised version merits your approval and satisfies the high standards of Nature Communications.

Response to Reviewers and a Summary of Changes

Many thanks to the reviewers for having given us valuable comments on the manuscript (NCOMMS-25-47700-A) submitted to *Nature Communications*.

Title: Spin hardening-boosted coherent magnon-exciton coupling in van der Waals heterostructure

Authors: Weican Lan, Chaocheng Liu, Ruiqi Liu, Yafei Chu, Lu Cheng, Yajuan Feng, Chao Wang, Huijuan Wang, Minghui Fan, Yuran Niu, Hengli Duan, Wensheng Yan

The reviewers presented professional comments and revision suggestions before the paper could be reconsidered for publication in *Nature Communications*. Their insights have been invaluable in guiding us towards enhancing the quality and clarity of our manuscript. We have seriously considered these suggestions and comments, made the revision accordingly, and would like to resubmit the revised manuscript. We sincerely appreciate the hard work of the reviewers and hope that the revised manuscript can meet the acceptance criteria. The detailed responses to the comments are presented in a point-to-point manner as follows. Accordingly, all responses to reviewers' comments are highlighted in **blue**; all revisions in the manuscript and supplementary files are highlighted in **red**.

Response to Reviewer #3

Lan et al have resubmitted their manuscript on exciton coupling to magnons in a CrSBr/Fe₃GaTe₂ heterostructure. They have sufficiently addressed all my technical comments, but I remain concerned about the overall impact of their results and whether their claims are overstated given the very small effect they report. They still have hyperbolic language in the text: "a breakthrough observation", etc. which seems to me unjustified. They have failed to explain why a 6 meV shift in a 1.3 eV feature (a 0.5% effect) that is 100 meV wide is enough to constitute a breakthrough or be practically useful for spintronics. I can imagine that if this effect could be enhanced this would be useful--but as it is, these results on their own seem neither particularly surprising or impressive.

On that note, I must insist on my previous request: that the authors explicitly state in the abstract and conclusion the percent modulation of the exciton energies they observe relative to the total bandwidth. The authors stated instead the percent of the shift of magnetic field, which is not what I asked. The casual reader who glances through just the abstract and conclusion needs to know exactly how large this effect is. Transparency about the magnitude is of utmost importance.

These results warrant publication somewhere, but unless the authors can clearly explain why their effect is as useful and breakthrough as they claim, it is perhaps not appropriate for the wide readership of Nature Communications.

Response: We sincerely thank the reviewer for the careful re-evaluation and constructive comments on our revised manuscript. We appreciate the reviewer's acknowledgment that our technical responses were satisfactory, and we have carefully considered the remaining concerns regarding the modulation magnitude of exciton energy and the significance of our results. In addition, any hyperbolic language that appears in the text has been removed in accordance with your rigorous consideration.

We apologize for the oversight of not explicitly stating the percent modulation of the exciton energies relative to the total bandwidth. We are grateful for your reminder. According to your suggestion, we have explicitly stated this modulation amplitude and added it to both the abstract and conclusion in our revised manuscript. We fully agree with your viewpoint that transparency about this modulation magnitude is crucial for directly highlighting the exciton tuning effect. Accordingly, we have extracted and analyzed the temperature-dependent PL data. Relative to the excitonic linewidth of approximately 100 meV, we achieved a bidirectional modulation of about 8.6% (redshift) and 6.1% (blueshift), as presented in **Fig. R1**. These shifts represent not only the movement of the PL peak, but also that of the entire PL spectrum. Notably, although

the absolute modulation magnitude in this work is relatively limited, the exciton energy shift can switch between positive and negative states, enabling bidirectional modulation. Specifically, relative to pristine CrSBr, we not only observe the generally reported redshift of the exciton energy but also discover an anomalous blueshift for the CrSBr in heterostructure. That is, we have achieved bidirectional modulation of the exciton energy, which has not been accomplished in previous studies to the best of our knowledge. We believe that this bidirectional capability is of greater significance than the magnitude alone, as it provides substantial flexibility in device design and paves the way for potential wavelength control in quantum information and optoelectronic technologies.

Moreover, we would like to emphasize once again that the value and innovation of our work are mainly in two aspects: (1) The modulation scheme is extended from conventional unidirectional control to bidirectional tunability (i.e., the exciton energy can be both red- and blue-shifted). This established a potential pathway for flexible exciton energy modulation on demand. (2) It demonstrates that exciton energy modulation can be achieved purely through intrinsic interfacial interactions, without relying on externally applied magnetic or electric fields. This establishes a distinct route toward exciton control, which is desirable for developing compact and self-sustained opto-spintronic devices.

Finally, we sincerely thank you for your careful review and constructive comments. We hope that our responses and revisions have addressed all concerns. Our work demonstrates the unique capability of bidirectional exciton energy modulation, which falls within the field of exciton physics and opto-spintronic devices. We believe that this study meets the high standards of *Nature Communications* and is of broad interest to its readership.

Fig. R1. Temperature-dependent modulation of exciton energy relative to the excitonic linewidth. The relative energy shift, expressed as a percentage of the excitonic linewidth (~ 100 meV), is plotted for three representative temperatures.

Revision:

Accordingly, **Fig. R1** with the relevant caption has been added on page 7 of the Supplementary Materials, as **Fig. S6**.

The original description on page 1 of the Main Text “Here we present an effective strategy to modulate the exciton state of CrSBr via phonon-mediated magnon-exciton tripartite interactions in the CrSBr/Fe₃GaTe₂ heterostructure.” **has been changed to** “Here we achieve bidirectional modulation of the CrSBr exciton energy via phonon-mediated magnon-exciton tripartite interactions in a CrSBr/Fe₃GaTe₂ heterostructure. Compared with pristine CrSBr, the photoluminescence (PL) peaks in the heterostructure can exhibit blueshift and redshift corresponding to 6.1% and 8.6% of the total bandwidth, respectively.”.

The original description on page 2 of the Main Text “Our findings demonstrate an approach for modest modulation of exciton energy in 2D AFM semiconductors, which may provide insights for future exploration of wavelength control in quantum information and optoelectronic technologies such as communication and sensing.” **has been changed to** “Our findings demonstrate an approach for bidirectionally modulating exciton energy in 2D AFM semiconductors, which provides substantial flexibility in device design and offers an avenue for potential wavelength control in quantum information and optoelectronic technologies.”.

The original description on page 3 of the Main Text “Here, we report an unprecedented observation of coherent exciton coupling and orientation-enhanced electron spin in a 2D magnetic heterostructure formed by CrSBr and Fe₃GaTe₂ (FGT).” **has been changed to** “Here, we report coherent exciton coupling and orientation-enhanced electron spin in a 2D magnetic heterostructure formed by CrSBr and Fe₃GaTe₂ (FGT).”.

The original description on page 3 of the Main Text “Moreover, with the devotion of the coupled phonons, this interfacial coupling consequently leads to an anomalous increase in photon emission energy.” **has been changed to** “Moreover, this interfacial coupling achieves not only a general photon energy redshift but also an anomalous blueshift, attributed to the mediation of coupled phonons.”.

The original description on page 3 of the Main Text “Through magnetotransport tests, we demonstrate this orientation-enhanced spin ordering in CrSBr, and simultaneously uncover that this coupling effect is reciprocal, as it meanwhile strengthens the magnetic properties of FGT.” **has been changed to** “Through magnetotransport tests, we demonstrate the orientation-enhanced spin ordering in CrSBr, and simultaneously uncover that interfacial magnetic coupling is reciprocal, as it meanwhile strengthens the magnetic properties of FGT.”.

The original description on page 3 of the Main Text “Consequently, the intrinsic AFM features are well-preserved, and the exciton energy of CrSBr demonstrates the

capability for selective modulation that is dependent on spin order.” **has been changed to** “Consequently, the intrinsic AFM features are well-preserved, and the exciton energy of CrSBr demonstrates the capability for bidirectional modulation dependent on spin order.”.

The original description on page 4 of the Main Text “Our work provides a novel approach for designing coherent behavior between magnons and excitons, laying the foundation for the application of a class of AFM semiconductors, exemplified by CrSBr, in the field of opto-spintronics.” **has been changed to** “Our work provides an approach for designing coherent behavior between magnons and excitons, offering new insights into the potential application of AFM semiconductors, exemplified by CrSBr, in the field of opto-spintronics.”.

The original description on page 6 of the Main Text “Furthermore, we observe that the temperature-dependent PL spectra exhibit higher photon energy below the T_N for the CrSBr on FGT, compared to pristine CrSBr (Fig. 2c), suggesting a correlation with its AFM phase transition under the action of FGT.” **has been changed to** “Furthermore, we observe that the temperature-dependent PL spectra exhibit higher photon energy (blueshift) below the T_N for the CrSBr on FGT, compared to pristine CrSBr (Fig. 2c), suggesting a correlation with its AFM phase transition under the action of FGT.”.

The original description on page 6 of the Main Text “Within these temperature ranges, CrSBr/FGT exhibits even lower photon energies compared to pristine CrSBr, achieving a maximum difference of around 200 K.” **has been changed to** “Within these temperature ranges, CrSBr/FGT exhibits even lower photon energies (redshift) compared to pristine CrSBr, achieving a maximum difference of around 200 K.”.

The original description on page 6 of the Main Text “A step height of ± 14 meV in exciton energy is acquired across critical temperature T_N , with its magnitude being comparable to the theoretical energy threshold under magnetic order modulation via external magnetic fields.” **has been changed to** “Compared with pristine CrSBr, a step height greater than 14 meV in photon energy is observed across the critical temperature (T_N) in the heterostructure, with a bidirectional modulation magnitude of approximately 6.1% (blueshift) and 8.6% (redshift) relative to the total excitonic linewidth of ~ 100 meV (Fig. S6).”.

The original description on page 14 of the Main Text “The tunable exciton energy between positive and negative states, dependent on magnon-exciton coupling, may unlock groundbreaking potential for arbitrary manipulation of exciton energies to enable precise control over the quantum states of excitons in quantum information storage.” **has been changed to** “The bidirectional tunable exciton energy between positive and negative states, dependent on magnon-exciton coupling, may unlock the potential for arbitrary manipulation of exciton energies to enable precise control over the quantum states of excitons in quantum information storage.”.

The original description on page **15** of the Main Text “In summary, we report a breakthrough observation of coherent exciton coupling and orientation-enhanced electron spin dynamics in a 2D CrSBr/FGT magnetic heterostructure.” **has been changed to** “In summary, we report coherent exciton coupling and orientation-enhanced electron spin dynamics in a 2D CrSBr/FGT magnetic heterostructure.”.

The original description on page **15** of the Main Text “Moreover, the coherently coupled exciton, which is affected by phonon-intervened magnon propagation as well, leads to an anomalous photon emission energy increase below T_N .” **has been changed to** “Moreover, the coherently coupled exciton, which is affected by phonon-intervened magnon propagation as well, leads to an anomalous photon emission energy blueshift below T_N .”.

The original description on page **15** of the Main Text “Notably, the exciton energy shift achieved in our work can reach 40% of the modulation amplitude achievable with external magnetic fields^[32,35], indicating a substantial level of modulation and offering a valuable reference for future studies on wavelength control in laser systems.” **has been changed to** “Notably, the exciton energy shift can be bidirectionally modulated between positive and negative states relative to pristine CrSBr, with the magnitude of blueshift and redshift exceeding 6% and 8% of the total bandwidth, respectively. This bidirectional control highlights significant flexibility in device design and offers a valuable reference for future studies on wavelength control in laser systems.”.

The original description on page **15** of the Main Text “Our findings advance a powerful strategy to tailor magnon-exciton coherence, opening avenues for AFM semiconductors like CrSBr in opto-spintronic devices such as spin-logic gates and nonvolatile magneto-optical memory systems.” **has been changed to** “Our findings present an effective strategy to tailor magnon-exciton coherence, opening avenues for AFM semiconductors like CrSBr in opto-spintronic devices such as spin-logic gates and nonvolatile magneto-optical memory systems.”.

Response to Reviewers and a Summary of Changes

Many thanks to the reviewers for having given us valuable comments on the manuscript (NCOMMS-25-47700-B) submitted to *Nature Communications*.

Title: Spin hardening-boosted coherent magnon-exciton coupling in van der Waals heterostructure

Authors: Weican Lan, Chaocheng Liu, Ruiqi Liu, Yafei Chu, Lu Cheng, Yajuan Feng, Chao Wang, Huijuan Wang, Minghui Fan, Yuran Niu, Hengli Duan, Wensheng Yan

The reviewers presented professional comments and revision suggestions before the paper could be reconsidered for publication in *Nature Communications*. Their insights have been invaluable in guiding us towards enhancing the quality and clarity of our manuscript. We have seriously considered these suggestions and comments, made the revision accordingly, and would like to resubmit the revised manuscript. We sincerely appreciate the hard work of the reviewers and hope that the revised manuscript can meet the acceptance criteria. The detailed responses to the comments are presented in a point-to-point manner as follows. Accordingly, all responses to reviewers' comments are highlighted in **blue**; all revisions in the manuscript and supplementary files are highlighted in **red**.

All revisions, along with the supplementary results and discussion (including those from the XMCD-PEEM measurements, DFT calculations, and micromagnetic simulations), have been incorporated into the revised paper.

Response to Reviewer #4

The manuscript entitled "Spin hardening-boosted coherent magnon-exciton coupling in van der Waals heterostructure" represents an impressively wide collection of experimental results comparing the properties of CrSBr and CrSBr/FGT heterostructures. The experiments are of high quality. However, I agree with previous referees that the broad implications of the results are overstated, and I find that the totality of the results do not seem to come together into a coherent story.

We sincerely appreciate your constructive comments on our manuscript. Regarding your concern about the potential overstatement of the broader implications of our results, we would like to emphasize that this issue had already been thoroughly reviewed and addressed during the previous round of revisions. We had carefully revised the entire manuscript in response to similar comments raised earlier, and these revisions were acknowledged and approved by Reviewer #3. As noted in his/her most recent report: *"They have sufficiently addressed all my technical comments."* and *"I thank the authors for putting the shift relative to total bandwidth in the abstract. I now am satisfied that my previous concerns have been met."* Accordingly, Reviewer #3 agreed that the manuscript is suitable for publication.

To ensure a coherent narrative and logical progression, our study was structured around a **motivation-observation-analysis-verification** framework. We began by building the CrSBr/FGT heterostructure, motivated by the potential of interfacial magnetic interactions to modulate excitonic properties, and characterized its structure and basic physical properties (Fig. 1). We then employed PL measurements to reveal how the interfacial coupling influences the optical response of CrSBr, from which we inferred a coupling relationship between magnetic order and excitons (Fig. 2). Subsequently, Hall transport measurements (Fig. 3) were carried out to examine how this coupling affects spin orientation, confirming an interface-induced enhancement of spin alignment (i.e., enhanced magnetic anisotropy). Finally, we further validated the interfacial magnetic interaction and clarified the fundamental driving mechanism behind the exciton energy shift through XAS, PEEM, as well as newly added theoretical calculations and micromagnetic simulations (Figs.4-5). These results collectively establish and refine the **charge-transfer-assisted** interfacial magnetic coupling mechanism on exciton energy shift.

Taken together, the overall structure and logical sequence of our work form a

coherent, self-consistent, and closed-loop narrative from experimental observations to microscopic mechanism verification.

The first main deficiency of the paper is that it asserts a lot of mechanisms for the observed properties without distinguishing those mechanisms from other possibilities or demonstrating consistency with the experiments through e.g. the use of simulations or models. Unless it can be made precise what their mechanisms are predicting, it is not clear that they are correct.

The second main deficiency is that many terms are ill-defined (e.g. spin-hardening, tripartite interaction), and ultimately come across more as a phenomenological description of the experimental data rather than a clearly defined microscopic explanation. Sometimes different parts of the manuscript seem to be referring to different mechanisms for the same observation, which further adds to the confusion.

The third deficiency is the overall results are underwhelming: the main conclusion that a small shift of the exciton PL, increase in saturation field, etc. can be induced by interfacial effects is not surprising.

Response: Thank you very much for your thoughtful comments and careful evaluation of our work.

First, according to your suggestions, we have supplemented the manuscript with additional experiments, together with comprehensive simulations and first-principles calculations, which together directly validate our experimental observations. These combined experimental and theoretical results allow us to clarify, from a microscopic perspective, the physical origin of the **interface magnetic coupling driven by charge transfer**, thereby providing more solid and comprehensive evidence for the overall physical picture: the charge transfer introduces additional spin polarization in CrSBr, thereby increasing its spin density and enhancing the interfacial magnetic interaction. Through their cooperative effect, both the exchange interaction strength and the magnetic anisotropy of CrSBr are enhanced. When CrSBr remains in the antiferromagnetic phase, the strengthened exchange interaction and magnetic anisotropy further stabilize the intralayer spin configuration, thereby suppressing interlayer charge hopping and the participation of interlayer excitons. This suppression reduces the contribution of interlayer electron-hole hybridization to optical transitions, ultimately manifesting as a blue shift of the exciton emission.

Specifically, our key conclusions are as follows:

1. Charge transfer occurs at the CrSBr/FGT interface

The occurred charge transfer manifests as a redistribution of magnetically charges associated with magnetic ordering (Fig. R1 and Table R1).

2. The identified interface magnetic coupling originates from charge transfer

The redistribution of charges at the interface modifies local electronic occupation and tunes the degree of orbital hybridization and overlap, thereby providing the essential electronic-structure conditions for interface exchange interactions (including direct exchange, superexchange, and RKKY-type coupling). In other words, the charge transfer between CrSBr and FGT alters the filling of interfacial *d* orbitals and the strength of their hybridization, forming the microscopic basis for enhanced interface magnetic exchange and magnetic anisotropy. This mechanism—“charge redistribution-orbital hybridization-enhanced magnetic exchange/coupling”—has been well established in numerous theoretical and experimental studies [*Phys. Rev. Applied* 12, 044010 (2019); *Adv. Sci.* 9, 2200186 (2022); *ACS Nano* 18, 12, 9232-9241 (2024); *Rev. Mod. Phys.* 89, 025006 (2017); *J. Am. Chem. Soc.* 146, 49, 34070-34079 (2024)], all of which demonstrate that interfacial charge transfer is a key driving force for tuning exchange interactions and interface magnetism. Notably, we have directly confirmed the existence of interface magnetic coupling by using DFT calculations and XMCD-PEEM tests (Figs. R1 and R2).

3. The enhanced spin alignment of CrSBr along the magnetic easy axis

Grazing-incidence XPEEM measurements first reveal that the magnetic domain configuration of CrSBr (Fig. R2) is strongly influenced by the underlying FGT layer, providing direct experimental evidence for an interfacial magnetic interaction between the two materials. To further elucidate the nature of this coupling, we performed micromagnetic simulations. Micromagnetic simulation results (Fig. R3) show that both the magnetic anisotropy and the exchange interaction strength of CrSBr in the heterostructure are increased, which is highly consistent with our observations of stable spin orientation in XPEEM and the enlarged coercive field in transport measurements. Meanwhile, first-principles calculations indicate an increase in the spin polarization rate difference of CrSBr (quantified by the asymmetry in spin-resolved charge density) (Fig. R4), which further reinforces its magnetic stability through enhanced spin-dependent Coulomb interactions.

4. The effect of enhanced antiferromagnetism on exciton energy

Most existing studies on tuning the exciton energy of CrSBr rely on applying an external magnetic field to drive the system from an interlayer antiferromagnetic (AFM) state into a ferromagnetic (FM) state. The underlying mechanism is that, in the AFM phase, spins in adjacent layers are antiparallel, which strongly suppresses interlayer electron-hole recombination. When an external magnetic field aligns the spins ferromagnetically, spin-allowed interlayer charge transfer channels are opened, leading to enhanced interlayer electron-hole hybridization and a redshift of the exciton energy (see Fig. R5, reproduced from [*Nat. Mater.* 20, 1657-1662 (2021); *Phys. Rev. B* 111, 075107 (2025)]).

In contrast, in our CrSBr/FGT heterostructure, the exchange interaction and magnetic anisotropy of CrSBr are enhanced by interfacial effects, which further stabilize the intralayer spin configuration and suppress interlayer charge hopping and interlayer exciton participation. As a result, the contribution of interlayer electron-hole hybridization to optical transitions is reduced, ultimately giving rise to the observed blueshift of the excitonic emission (Fig. R6).

Based on the above analyses, the mechanism we now propose is fully self-consistent and explains the two central observations in the CrSBr/FGT heterostructure:

- (1) the enhanced antiferromagnetic order of CrSBr, manifested by increased exchange interaction strength and magnetic anisotropy;**
- (2) the anomalous blueshift of the CrSBr exciton energy.**

Overall, we have systematically reorganized the discussion of the physical mechanism in the revised manuscript. The description of the magnetic-exciton coupling driven by interface magnetic interactions is now clearer, logically consistent, and well aligned with both experimental and theoretical evidence. All relevant content has been thoroughly updated and clarified in the main text.

Second, we sincerely acknowledge that some descriptions regarding the underlying mechanisms in the previous version were not sufficiently clear or rigorous, which may have caused potential misunderstanding or confusion for readers. In response to your comments, we have systematically revised the relevant sections in the new manuscript: we removed terms such as “*spin hardening*” and “*coherent magnon-exciton coupling*”, which may lead to ambiguity, and consistently adopted more physically accurate and experimentally supported terminologies—namely “*enhanced magnetic anisotropy*” and “*spin-exciton coupling*.” In addition, based on the experimental data, we have thoroughly ruled out alternative factors such as strain and thermal expansion that could

otherwise influence the exciton energy (see detailed analyses in the revised manuscript).

Finally, regarding the shift of the exciton PL peak, we would like to emphasize once again that this issue has already been addressed in detail in our previous response. For example, we previously stated: *“Relative to pristine CrSBr, we not only observe the generally reported redshift of the exciton energy but also discover an anomalous blueshift for CrSBr in the heterostructure. That is, we have achieved bidirectional modulation of the exciton energy, which, to the best of our knowledge, has not been realized in previous studies. We believe that this bidirectional capability is of greater significance than the magnitude alone, as it provides substantial flexibility in device design and opens pathways toward potential wavelength control in quantum information and optoelectronic technologies.”* These points have now been explicitly included in the revised manuscript. It is worth noting that the magnitude of the exciton energy shift in our system is actually non-negligible in this research field, and more importantly, we achieve **bidirectional** modulation of the PL exciton peak (both redshift and blueshift), which is a key discovery. The uniqueness and validity of this bidirectional modulation have been clearly acknowledged by Reviewer #3, as reflected in their comment: *“I thank the authors for putting the shift relative to total bandwidth in the abstract. I now am satisfied that my previous concerns have been met.”*

Fig. R1. The charge density differences for CrSBr/FGT heterostructure. The pink and green areas depict charge accumulation and depletion, respectively.

Table R1. Amount of electron transfer. DFT calculated electronic transfer in CrSBr/FGT heterostructure. The symbol "+" denotes electron gain, while "-" represents electron loss. Unit: $e/\text{\AA}^2$.

CrSBr	FGT
+0.007	-0.007

Fig. R2. XMCD-PEEM images of Cr *L*-edge acquired with a 16° X-ray incidence angle relative to the sample surface. The red and green dashed lines mark the areas of CrSBr and FGT, respectively.

Fig. R3. Micromagnetic simulations based on the constructed heterostructure model and the corresponding extracted energies. (a) Schematic of the four-layer model, consisting of two out-of-plane ferromagnetic FGT layers and two in-plane antiferromagnetic FGT layers. (b) Micromagnetic simulation results of the exchange interaction and magnetic anisotropy energies of CrSBr in the pristine and heterostructure. Procedure for micromagnetic parameter extraction: first, the interlayer coupling in the heterostructure is turned off (i.e., FGT and CrSBr are treated as independent regions), after which the anisotropy and exchange energies are extracted for each layer. Then, the interlayer coupling is enabled, the anisotropy and exchange energies are re-evaluated for comparison.

Fig. R4. The spin polarization rate difference (ΔP) of CrSBr in the heterostructure, which can be expressed as $\Delta P = |(N_{up}^{hetero} - N_{down}^{hetero}) / N_{total}^{hetero}| - |(N_{up}^{CrSBr} - N_{down}^{CrSBr}) / N_{total}^{CrSBr}|$. where N_{up} and N_{down} denote the numbers of spin-up and spin-down electrons, respectively, and N_{tot} is the total number of electrons. The superscripts “hetero” and “CrSBr” refer to the CrSBr in the CrSBr/FGT heterostructure and pristine CrSBr, respectively. The red and blue areas represent positive and negative values of the spin polarization rate difference, respectively.

[FIGURE REDACTED]

Fig. R5 Schematic illustration of the effect of spin alignment on interlayer exciton behavior in CrSBr. (a) [*Nat. Mater.* 20, 1657-1662 (2021)]. (b) [*Phys. Rev. B* 111, 075107 (2025)]. In the AFM bilayer, interlayer antiferromagnetic order suppresses hybridization between layers, leading to the localization of spin-up and spin-down electrons at the top and bottom layers, respectively, near the doubly degenerate conduction band minima and valence band maximum. As a result, excitons are primarily intralayer, with the electron and hole confined to the same layer, as confirmed by the side view of the exciton wavefunction (Fig. a-upper panel). In contrast, in the FM bilayer, interlayer hybridization is allowed, so the electron associated with a hole fixed in one layer exhibits amplitude across both layers (Fig. a-lower panel), indicating enhanced interlayer electron-hole hybridization in the FM state, which leads

to a redshift of the exciton energy.

Fig. R6 Schematic illustration of the exciton recombination process in CrSBr under the enhanced antiferromagnetic configuration. The enhancement of antiferromagnetic exchange interaction and magnetic anisotropy stabilizes the layer-resolved spin configuration, which further suppresses spin-allowed interlayer charge transfer and interlayer exciton formation. As a result, exciton recombination is dominated by intralayer processes, giving rise to the observed blueshift of the exciton emission.

Revision:

Accordingly, **Fig. R1, R3b, R4, R6** with the relevant caption has been added on page **14** of the Main Text, as **Fig. 5**.

Accordingly, **Fig. R2** with the relevant caption has been added on page **12** of the Main Text, as **Fig. 4i**.

Accordingly, **Fig. R3a** with the relevant caption has been added on page **21** of the Supplementary Materials, as **Fig. S20**.

Accordingly, **Table. R1** with the relevant caption has been added on page **22** of the Supplementary Materials, as **Table. S1**.

The supplementary description “Lastly, we discuss the anomalous increase of exciton energy in CrSBr/FGT below T_N . To understand this phenomenon from a microscopic perspective, we performed a series of theoretical calculations and simulations. Density functional theory (DFT) calculations reveal a pronounced redistribution of charge density in the CrSBr/FGT heterostructure (**Fig. 5a**). Quantitative charge analysis (**Table S1**, Supporting Information) shows a net electron transfer of approximately $0.007 e/\text{\AA}^2$ from FGT to CrSBr, in agreement with our KPFM and XAS measurements, further confirming the existence of interfacial charge transfer. Notably, such charge transfer

occurring between magnetic materials provides a crucial electronic-structure foundation for the establishment of interfacial magnetic interaction. On this basis, we further extract the spin polarization rate difference (ΔP), of CrSBr in the heterostructure from the DFT calculations (**Fig. 5b**), which can be expressed as $\Delta P = |(N_{up}^{hetero} - N_{down}^{hetero}) / N_{total}^{hetero}| - |(N_{up}^{CrSBr} - N_{down}^{CrSBr}) / N_{total}^{CrSBr}|$, where N_{up} , N_{down} and N_{tot} denote the numbers of spin-up, spin-down and total electrons, respectively. The superscripts “hetero” and “CrSBr” refer to the CrSBr in the CrSBr/FGT heterostructure and pristine CrSBr, respectively. A positive ΔP indicates that the spin-polarized rate of CrSBr is enhanced in the heterostructure compared to pristine CrSBr, providing direct microscopic electronic-structure evidence for the modulated spin configuration of CrSBr. To further elucidate how these electronic-structure modifications influence the magnetic properties, we performed micromagnetic simulations based on parameters extracted from both experiments and theoretical calculations. Micromagnetic simulations based on MuMax3 serve as a powerful tool for capturing the evolution of magnetic energies and spin configurations in magnetic materials, particularly for our constructed heterostructure model incorporating interlayer magnetic coupling between FGT and CrSBr (**Fig. S20**). The simulation results demonstrate that both the magnetic anisotropy energy and the exchange interaction energy of CrSBr are enhanced in the heterostructure (**Fig. 5c**), which is consistent with the emergence of magnetic single-domain of CrSBr observed in the XMCD-PEEM measurement. This elevated magnetic energy landscape effectively stabilizes the AFM spin configuration of CrSBr, which is in excellent agreement with the robust spin orientation in XMLD-PEEM and the increased spin flip field in transport measurements, and is further supported by the enhanced spin polarization rate revealed by the DFT calculations.

Combining the above experimental observations, micromagnetic simulations, and DFT calculations, we establish the following physical picture: interfacial charge-transfer-driven magnetic coupling enhances the AFM ordering of CrSBr, manifested by the concurrent enhancement of its exchange interaction and magnetic anisotropy. It is worth noting that for A-type AFM CrSBr, antiparallel spin alignment between adjacent layers strongly suppresses interlayer electron-hole recombination. When an external magnetic field or other perturbations drive the interlayer spins into a ferromagnetically aligned configuration, spin-allowed interlayer charge transfer channels are opened, leading to enhanced interlayer electron-hole hybridization and a redshift of the exciton energy. In contrast, in our CrSBr/FGT heterostructure, the interfacial magnetic interaction further strengthens the AFM order of CrSBr, thereby more effectively

suppressing interlayer electron-hole recombination and significantly reducing its contribution to optical transitions (**Fig. 5d**). Our work ultimately gives rise to an anomalous blueshift of the exciton emission energy below T_N through interfacial magnetic interaction. When the temperature exceeds the T_N of CrSBr, its AFM order gradually diminishes, and the interfacial magnetic interaction-modulated ability correspondingly weakens, resulting in an abrupt change in the PL energy across the Néel temperature (**Fig. 2d**). In the temperature range above the T_N of CrSBr but below the T_C of FGT, ferromagnetically ordered FGT may still locally induce FM correlations in CrSBr, which can reopen interlayer electronic hybridization channels and lead to a redshift of the exciton energy. As the temperature is further increased beyond the T_C of FGT (above 300 K), this induced effect vanishes, and the exciton energy in the CrSBr/FGT heterostructure gradually converges to that of pristine CrSBr. This tunable exciton energy between blueshift and redshift, driven by interfacial magnetic interaction-dependent spin-exciton coupling, may unlock the potential for arbitrary manipulation of exciton energies to enable precise control over the quantum states of excitons in quantum information storage.” **has been added** on page **13** of the Main Text.

The original description on page **1** of the abstract “Here we achieve bidirectional modulation of the CrSBr exciton energy via phonon-mediated magnon-exciton tripartite interactions in a CrSBr/Fe₃GaTe₂ heterostructure. Compared with pristine CrSBr, the photoluminescence (PL) peaks in the heterostructure can exhibit blueshift and redshift corresponding to 6.1% and 8.6% of the total bandwidth, respectively. We demonstrate a reciprocal spin-hardening behavior that is induced by interfacial magnetic coupling and orbital-mediated charge transfer and that suppresses the phonon vibrations of magnetic ions.” **has been changed to** “Here we achieve bidirectional modulation of the CrSBr exciton energy via interfacial interaction-modified spin-exciton coupling in a CrSBr/Fe₃GaTe₂(FGT) heterostructure. Compared with pristine CrSBr, the photoluminescence (PL) peaks in the heterostructure can exhibit blueshift and redshift corresponding to 6.1% and 8.6% of the total bandwidth, respectively. We reveal that the interfacial charge-transfer-driven magnetic coupling in the heterostructure effectively enhances the magnetic anisotropy and the exchange interaction of CrSBr, thereby stabilizing its AFM spin configuration, suppressing interlayer electron-hole recombination, and ultimately leading to an anomalous blueshift of the exciton emission.”.

The original description on page **3** of the introduction “Here, we report coherent exciton coupling and orientation-enhanced electron spin in a 2D magnetic

heterostructure formed by CrSBr and Fe₃GaTe₂ (FGT). The features of phonon modes and exciton PL in CrSBr exhibit a synergistic dependence on the magnetic transition temperatures of both CrSBr and FGT, suggesting strong tripartite coupling among phonons, magnons, and excitons. Due to the robust interfacial magnetic exchange interactions imparted by FGT, the easy-axis AFM spin ordering in CrSBr is hardened and reinforced orientationally, altering the vibrational polarity of the magnetic ions. Moreover, this interfacial coupling achieves not only a general photon energy redshift but also an anomalous blueshift, attributed to the mediation of coupled phonons. Through magnetotransport tests, we demonstrate the orientation-enhanced spin ordering in CrSBr, and simultaneously uncover that interfacial magnetic coupling is reciprocal, as it meanwhile strengthens the magnetic properties of FGT. Furthermore, in combination with microscopic-level synchrotron X-ray photoemission electron microscopy (XPEEM), we reveal an orbital-mediated charge transfer mechanism (from Fe e_g -orbital to Cr t_{2g} -orbital) for stabilizing spin order (spin hardening), without altering the magnetic domain structure of either FGT or CrSBr. Consequently, the intrinsic AFM features are well-preserved, and the exciton energy of CrSBr demonstrates the capability for bidirectional modulation dependent on spin order.” **has been changed to** “Here, we report interfacial magnetic interaction-modified spin-exciton coupling in a 2D magnetic heterostructure formed by CrSBr and FGT. The features of exciton PL in CrSBr exhibit a tight dependence on the magnetic transition temperatures of both CrSBr and FGT, suggesting the existence of spin-exciton coupling. In contrast to the common PL redshift in CrSBr, we demonstrate that the exciton energy exhibits an anomalous blueshift, originating from the suppression of interlayer electron-hole recombination induced by interfacial magnetic interaction. Through magnetotransport tests, we demonstrate the magnetic properties of both FGT and CrSBr are modulated by the interfacial interaction. Furthermore, in combination with microscopic-level synchrotron X-ray photoemission electron microscopy (XPEEM), we reveal an orbital-mediated charge transfer mechanism (from Fe e_g -orbital to Cr t_{2g} -orbital) for the interfacial magnetic interaction between CrSBr and FGT, which alters the magnetic domain of CrSBr. Micromagnetic simulations combined with theoretical calculations further support that the observed excitonic blueshift originates from the suppression of interlayer electron-hole recombination driven by interfacial magnetic interactions.”.

The original description on page 15 of the conclusion “In summary, we report coherent exciton coupling and orientation-enhanced electron spin dynamics in a 2D CrSBr/FGT magnetic heterostructure. Under the effect of FM order in FGT, the phonon modes and excitonic PL in CrSBr exhibit strong correlations with the magnetic phase transition, demonstrating spin-phonon and magnon-exciton coupling. Moreover, the

coherently coupled exciton, which is affected by phonon-intervened magnon propagation as well, leads to an anomalous photon emission energy blueshift below T_N . We demonstrate that the robust interfacial magnetic exchange interactions from FGT harden the AFM spin ordering in CrSBr, which therefore modifies the vibrational polarity of magnetic ions and elevates the spin-flop field. Both magnetotransport and synchrotron XPEEM confirm that this interfacial coupling, accompanied by a charge transfer-mediated mechanism, not only orientationally enhances the spin ordering in CrSBr but also reciprocally amplifies the magnetization of FGT, all while preserving their intrinsic magnetic domain structures.” **has been changed to** “In summary, we demonstrate the bidirectional tunability of exciton energy in a 2D CrSBr/FGT magnetic heterostructure, governed by spin-exciton coupling modified through interfacial magnetic interactions. The FM order in FGT modulates the spin configuration of CrSBr, which is consistently reflected in the evolution of magnetic-sensitive exciton energy, leading to an anomalous photon emission energy blueshift below T_N . These results establish a direct correlation between interfacial spin states and exciton energy, highlighting spin-mediated control of excitonic properties in vdW heterostructures. Based on experimental measurements and theoretical simulations, we demonstrate that the interfacial charge-transfer-driven magnetic coupling in the heterostructure enhances both the magnetic anisotropy and exchange interaction of CrSBr, effectively stabilizing its AFM spin configuration, which suppresses interlayer electron-hole recombination and increases the spin-flip field. ”.

The supplementary description “This enhanced magnetoresistance may also influence the interlayer electronic interactions, potentially affecting the excitonic behavior.” **has been added** on page 8 of the Main Text.

The original description on page 8 of the Main Text “All these results declare that the interfacial magnetic coupling in the CrSBr/FGT heterostructure is reciprocal, as it strengthens the alignment of spin order for both CrSBr and FGT (spin hardening), and affects their electrical properties.” **has been changed to** “All these results declare that the interfacial magnetic interaction in the CrSBr/FGT heterostructure is reciprocal, which affects the magnetic properties of both CrSBr and FGT. ”.

The original description on page 10 of the Main Text “All XPEEM data were recorded after cooling from room temperature to 100 K under zero magnetic field with normal incidence (90°) of X-ray on the sample surface. This setup is sensitive to out-of-plane FM and in-plane AFM but cannot detect the in-plane FM domain.” **has been changed to** “All XPEEM data were recorded at 100 K after zero-field cooling with

normal incidence (90°) of X-ray on the sample surface. Exceptionally, **Fig. 4i** was measured using grazing-incidence of X-ray (16°). ”.

The supplementary description “To directly probe the in-plane magnetic domains of CrSBr and validate the interfacial interaction in the heterostructure, we further performed Cr *L*-edge XMCD-PEEM measurements under grazing-incidence geometry (16°) (**Fig. 4i**), which enhances sensitivity to the in-plane magnetic component of Cr moments. Notably, compared with the multidomain state of pristine CrSBr, CrSBr in the heterostructure exhibits a magnetic single-domain with decreased magnetic contrast. This direct observation clearly indicates that the enhanced magnetic domain wall energy and decreased magnetic moment for CrSBr in CrSBr/FGT heterostructure, further demonstrating the presence of interfacial magnetic interaction between CrSBr and FGT.” **has been added** on page **10** of the Main Text.

The original description on page **11** of the Main Text “Based on the above discussion, although the magnetic coupling interactions in the CrSBr/FGT heterostructure enhance spin alignment to achieve a spin-hardened state, it does not alter their intrinsic AFM spin orientation or magnetic domain configuration.” **has been changed to** “Based on the above discussion, although the interfacial magnetic interaction in the CrSBr/FGT heterostructure changes the magnetic domain of CrSBr, it does not alter the intrinsic AFM spin orientation. ”.

The original description on page **11** of the Main Text “Specifically, the increased electron occupancy in Cr-3*d* orbitals of CrSBr enhances their electron cloud density and amplifies the spatial overlap of adjacent Cr atomic orbitals. As a result, the strengthened orbital hybridization and Cr-Cr AFM coupling further stabilize the AFM spins along the zigzag chains and enhance the magnetic proximity interactions between CrSBr and FGT, collectively contributing to the spin hardening and the suppressed lattice vibration.” **has been changed to** “Specifically, the increased electron occupancy in Cr-3*d* orbitals of CrSBr enhances their electron cloud density and amplifies the spatial overlap of adjacent Cr atomic orbitals, which in turn strengthens orbital hybridization, stabilizes the spin configuration, and reinforces the interfacial magnetic interaction. ”.

1. What is the meaning of "phonon-mediated magnon-exciton tripartite interactions"?

It seems the temperature dependence of the PL and Raman data in Fig. 2 can be attributed to regular thermal expansion and magneto-elastic coupling. It is clear that the exciton energy may be influenced by the local crystalline environment; Cr(III) complexes display a wide range of PL emission energies ... Cr(III) PL in rubies is of course famously used for in-situ pressure sensors. It is clear the structure has a temperature dependence, through anharmonicity and in response to the magnetic correlations. These effects would be expected to be modified in the heterostructures due to differences in thermal expansivity and magneto-elastic coupling in combination with interfacial strain or charge transfer, so that CrSBr and CrSBr/FGT would be expected to display slightly different temperature dependence of the PL and phonon Raman response. I do not find the observations surprising.

Is the author's scenario the same or different from these conventional effects?

If it is the same, there is no need for a new name for these effects.

If it is different, the authors should indicate how it differs from conventional effects, present theory showing how it may be distinguished, and demonstrate it experimentally.

Response: We thank the reviewer for the careful evaluation of the phenomena presented in Fig. 2. We fully agree that conventional effects-such as thermal expansion, magnetoelastic coupling, interfacial strain, and charge transfer can simultaneously influence the temperature evolution of PL and Raman spectra in heterostructure systems. In the revised manuscript, we have systematically examined and ruled out these possibilities one by one. Meanwhile, we acknowledge that the current experimental evidence is insufficient to directly establish a phonon-mediated role in the magnon-exciton interaction (as we previously described as “phonon-mediated magnon-exciton tripartite interactions”). The PL and Raman temperature dependences primarily reflect the influence of interfacial interactions on the exciton energy, rather than a direct phononic contribution. Therefore, we have removed this statement.

To further clarify the dominant mechanism, we have substantially expanded the revised manuscript by incorporating PEEM measurements (Fig. R2), first-principles DFT calculations (Fig. R1 and Fig. 5 in main text), and micromagnetic simulations (Fig.

5 in main text), which are jointly analyzed alongside the experimental data. These analyses consistently point to an interface magnetic interaction driven by charge transfer as the physically grounded mechanism that underlies the observed enhancement of antiferromagnetism and the anomalous blueshift of the exciton energy. The main viewpoints and evidence are presented below:

1. Distinct temperature-dependent behavior from conventional thermal effects

Thermal expansion or electron-phonon coupling generally leads to a continuous and monotonic redshift of the exciton energy across the entire temperature range, a phenomenon widely reported in semiconductors [*Appl. Phys. Lett.* 58, 2924-2926 (1991); *Nat. Phys.* 14, 801-805 (2018); *Nat. Mater.* 20, 1657-1662 (2021)]. In contrast, the key feature of our heterostructure is that **the exciton blueshift appears only below the Néel temperature (T_N) of CrSBr**, accompanied by a marked enhancement of the switching field. This “kink” around T_N is fundamentally distinct from the smooth, linear thermal redshift and therefore cannot be explained by thermal effects alone.

2. Exclusion of interfacial strain

For FGT: Previous studies show that strain induces opposite trends in the A1 and E2 Raman modes of FGT [*Adv. Funct. Mater.* 34, 2400552 (2024); *Adv. Sci.* 10, 2206842 (2023)]. However, in our heterostructure, the Raman peaks of FGT (Fig. 1e) do not exhibit this signature. Instead, their shifts are consistent with overall charge-transfer effects.

For CrSBr: Strain in Cr-based layered materials typically alters lattice constants or orbital anisotropy, resulting in measurable changes in the XMLD line shape and peak positions [*Phys. Rev. B* 82, 184415 (2010); *Phys. Rev. B* 105, 134416 (2022)]. However, our XMLD measurements show neither peak shifts nor profile changes, indicating the absence of strain-induced effects. We also emphasize that Cr(III) in rubies is structurally and chemically distinct from CrSBr; its strain response cannot be generalized to our system.

For sample fabrication: Our heterostructures were assembled by mechanical exfoliation and dry transfer rather than epitaxial growth. Therefore, there is no lattice-mismatch-induced epitaxial strain. Residual mechanical stress introduced during transfer was significantly reduced by high-vacuum annealing.

Taken together, Raman evidence, XMLD results, and the fabrication/annealing process consistently demonstrate the absence of significant interfacial strain or strain-induced magnetoelastic coupling.

3. Charge transfer exists but is not the direct cause of the anomalous blueshift

KPFM/XAS measurements show that charge transfer persists across the entire temperature range, whereas the exciton blueshift appears only below T_N . Thus, charge transfer alone cannot account for the temperature-selective behavior.

Instead, as discussed earlier, charge transfer more likely serves as a **precondition**, modulating orbital occupation and strengthening interfacial exchange interactions, rather than directly shifting the exciton energy.

Control experiment: We previously performed a control experiment by replacing FGT with nonmagnetic metallic WTe₂, which has a conductivity comparable to FGT [*Nature* 514, 205-208 (2014); *Chin. Phys. Lett.* 40, 087501 (2023)]. KPFM still revealed charge transfer in the same direction (Fig. S8a). However, without interfacial magnetic interactions, the PL spectra show **no correlation** with the magnetic ordering of CrSBr (Figs. S8b,c). This clearly confirms that **charge transfer alone cannot produce the observed temperature-selective blueshift**.

4. Micromagnetic simulations and DFT calculations reveal the microscopic mechanism

Building on the direct experimental evidence from PEEM that the magnetic domains of CrSBr are influenced by the underlying FGT layer (Fig. R2), exhibiting a transition from a multidomain to a single-domain configuration and indicating the presence of an interfacial magnetic interaction. We further employed micromagnetic simulations and DFT calculations to uncover the microscopic origin of this coupling. We conducted micromagnetic simulations using Mumax3, which is a widely used GPU-accelerated micromagnetics platform based on the Landau-Lifshitz-Gilbert (LLG) equation and capable of treating exchange, anisotropy, Zeeman, and interface contributions [*AIP Adv.* 4, 107133 (2014)]. Using Mumax3, we constructed a CrSBr/FGT heterostructure model and extracted the anisotropy and exchange energies for pristine and coupled systems. Simulation results show (Fig. R3):

- **CrSBr exhibit enhanced magnetic anisotropy**, not only aligns with the elevated switching/coercive fields observed in transport measurements, but also provides a natural explanation for the single-domain magnetic configuration seen in PEEM.
- **CrSBr's exchange interaction energy increases**, owing to additional electron occupancy in the Cr-3d orbitals and the enhanced spin polarization difference, which increases orbital overlap and strengthens the effective exchange interaction.

Meanwhile, we note that the magnetic ordering in CrSBr significantly affects interlayer exciton recombination: when the system is in the **interlayer**

antiferromagnetic state, spins in adjacent layers are aligned antiparallel, which strongly suppresses spin-allowed interlayer electron-hole recombination; in contrast, when the magnetic order is tuned to an **interlayer ferromagnetic configuration**, the spin selection rules are relaxed, opening interlayer charge transfer and exciton recombination channels, thereby substantially affecting the exciton energy [*Nat. Mater.* 20, 1657-1662 (2021); *Phys. Rev. B* 111, 075107 (2025)].

Based on this understanding, we further performed density functional theory (DFT) calculations to obtain the **spin polarization rate difference** in CrSBr within the heterostructure (Fig. R4). The results show that, compared with intrinsic CrSBr, the spin polarization rate (i.e., the asymmetry in spin-resolved charge density) in heterostructured CrSBr is significantly enhanced, indicating an increase in its spin polarization. This finding is highly consistent with the experimentally observed enhancement of magnetic anisotropy and related physical properties.

5. Magnetic-order-dependent signatures provide cross-validation

Experimentally, we observe:

- (i) enhanced switching/saturation field in the heterostructure;
- (ii) a sudden PL energy change near the T_N of CrSBr;
- (iii) DFT evidence of charge transfer and increase of spin polarization rate.

These three independent but mutually consistent signatures strongly support the causal mechanism of **interface magnetic coupling-induced exciton blueshift**, ruling out mixed effects from thermal or strain contributions.

Taken together, our experimental observations, micromagnetic simulations, and first-principles calculations establish a complete and self-consistent mechanistic validation.

2. Is the "spin hardening" a consequence of exchange coupling between the FGT and the CrSBr, or a modification of the magnetic anisotropy (for example via charge transfer or strain)?

First, it should be noted that the magnetic anisotropy of FGT and related compounds is sensitive to both the Fe valency and strain (see discussion in Wu, S. et al Nature Communications 15, 10765 (2024)).

Second, the increase of the saturation field of CrSBr when in contact with FGT seems to scale linearly with the magnetic order parameter of CrSBr, which would be the expected result if the magnetic anisotropy of CrSBr were increased or exchange between CrSBr layers was increased. If it were the direct result of exchange coupling with the FGT, it would be expected to scale with the magnetization of the FGT, which it does not.

The discussion of this effect is confusing in the text.

For example, page 12 reads: "the interfacial charge transfer can simultaneously modify the electronic structure of Cr, thereby exerting a profound impact on macroscopic magnetic properties. Specifically, the increased electron occupancy in Cr-3d orbitals of CrSBr enhances their electron cloud density and amplifies the spatial overlap of adjacent Cr atomic orbitals[52-54]. As a result, the strengthened orbital hybridization and Cr-Cr AFM coupling further stabilize the AFM spins along the zigzag chains and enhance the magnetic proximity interactions between CrSBr and FGT, collectively contributing to the spin hardening and the suppressed lattice vibration."

Here it seems to imply that charge transfer results in a modification of the magnetic couplings within the CrSBr.

However, the conclusion reads "We demonstrate that the robust interfacial magnetic exchange interactions from FGT harden the AFM spin ordering in CrSBr, which therefore modifies the vibrational polarity of magnetic ions and elevates the spin-flop field."

Here it seems to imply magnetic exchange interactions between FGT and CrSBr, not charge transfer. What do the authors mean? What is meant by vibrational polarity?

Response: Thank you for your thoughtful and constructive feedback on our manuscript. To begin with, we have realized that the term “*spin hardening*” does not clearly convey the underlying physical mechanism and only describes the phenomenological outcome of interface-induced magnetic modification. We have therefore revised it to “*enhanced magnetic anisotropy*,” which more accurately captures the physics. This mechanism is corroborated by our micromagnetic simulations: the interfacial magnetic interaction in the CrSBr/FGT heterostructure significantly increases the magnetic anisotropy of CrSBr. As discussed earlier, charge transfer plays an assisting role in this process by providing the necessary electronic environment for interfacial magnetic coupling, rather than directly driving the exciton energy shift.

Regarding the strain in FGT, we have provided a detailed clarification in our response to Q1. In our heterostructure, strain is extremely small and can be negligible.

We fully agree with the reviewer’s point that “*the increase of the saturation field of CrSBr would be the expected result if the magnetic anisotropy of CrSBr were increased or if the exchange between CrSBr layers were strengthened.*” Indeed, both our theoretical calculations and micromagnetic simulations consistently show that charge-transfer-assisted interfacial magnetic coupling enhances both the exchange interaction and the magnetic anisotropy of CrSBr, which naturally leads to an increase in its switching field.

For the reviewer’s further comment that “*the saturation field of CrSBr would be expected to scale with the magnetization of FGT*” we offer the following clarification: the switching field of CrSBr is primarily determined by its **intrinsic magnetic anisotropy** and **magnetic ordering strength**, rather than directly by the magnetization amplitude of FGT. In our heterostructure, the switching field of CrSBr is governed by two intertwined factors:

- (1) the intrinsic magnetic order of CrSBr, which weakens with increasing temperature owing to thermal fluctuations and reduced AFM order stability;
- (2) the strength of interfacial magnetic coupling, which depends on the magnetic ordering of both materials.

Therefore, the switching field cannot be expected to scale directly with the magnetization of FGT alone, as it is determined by a combination of these above factors.

Regarding the structure-function relationship between charge transfer and interfacial magnetic coupling, we have clarified that the two are inherently unified: charge transfer provides the essential driving force for the emergence of interfacial magnetic exchange. The origin of interfacial magnetic interaction is Coulomb interaction, which typically involves the key process and essential role of charge transfer, broadly seen in many typical magnetic interactions, such as the RKKY and super-exchange interaction. Moreover, our theoretical calculations and micromagnetic modeling explicitly confirm that charge-transfer-induced interfacial magnetic coupling enhances the magnetic anisotropy of CrSBr, increases spin splitting in the band structure, and ultimately results in the observed exciton blueshift. Notably, in response to your feedback, we have further clarified the unified relationship between charge transfer and interfacial magnetic coupling in the revised manuscript.

Finally, regarding the term “*vibrational polarity*”, it originally referred to the characteristic vibrational pattern of Cr magnetic ions along the zigzag chain. We have now revised it to the more accurate term “*vibrational feature*” in accordance with your feedback.

3. I do not view the collected data as unique indication of coherent exciton-magnon coupling.

First, demonstrating that the PL is influenced by the magnetic state of the material is not sufficient to imply coherent exciton-magnon coupling; as noted above, the PL may be influenced by effects like strain and charge transfer, which are clearly implied by the other measurements. The fact that the structure (and therefore every other property) is sensitive to the magnetic ground state through magneto-elastic coupling is not the same as coherent coupling of the exciton to magnetic excitations above the ground state. Here is a place where an explicit model, with precise predictions for the temperature or field dependence, should have been employed to justify the uniqueness of the conclusions of the manuscript.

Second, the PL in CrSBr has previously been demonstrated to have significant vibrational fine structure (see Lin et al, ACS Nano 2024, 18, 2898, which should have been cited in the present work). Much of the observed lineshape is likely dominated by this fine structure. It is more accurate to say that the lineshape (including the large vibrational contribution) has been modified by FGT, but it is not clear the fundamental exciton has shifted. In Fig. 2c there are clearly differences in the lineshapes and linewidths, making the peak position itself a complicated measure of any effect.

Third, it must be remarked that the slope of the PL peak energy as a function of temperature remains similar for $T > T_N$ and $T < T_N$. However, for $T > T_N$ there are no coherent magnons to couple with. It is stated in the main text: "we perform a fitting analysis of the exciton peak shift using the power-law dependence of the AFM transition and yield a well-fitted result (Fig. S5), which further provides compelling evidence for coherent magnon-exciton coupling in the heterostructure." However, the power law fitting of the peak position in Fig. S5 is not very compelling at all. The data shows a continuous shift of the peak energy that is considerably more broad than the fitted function, suggesting there are mechanisms other than magnetic ordering contributing to the shift.

As an additional comment: the extraction of the peak energy itself does not appear consistent across different figures. What is the meaning of the black lines in Fig. 2(a,b)?

Why do they not match the peak positions plotted in Fig. 2d? Why do the positions of the arrows in Fig. 2c not correspond to the maxima of the curves, and also not match the peak positions in Fig. 2d? (e.g. the arrows at 250 and 200K in Fig. 2c appear to be at the same energy, but they are not in Fig. 2d).

Also: the caption of Fig. S5 says that it was fit to a function with $I^{(2w)}$, but this is not second harmonic generation; it is PL peak position. It is not clear why $I^{(2w)}$ appears in this expression. The same expression appears in the caption of Fig. S12, which addresses Raman phonon peak intensity, which is also not $I^{(2w)}$.

Response: We appreciate the reviewer's careful assessment and fully understand the concern.

First, we agree with the reviewer that the current experimental data alone cannot uniquely demonstrate coherent magnon-exciton coupling. As pointed out, the PL response to magnetic states is not sufficient to establish coherent coupling. In response to this valuable comment, we have removed terms such as “coherent magnon-exciton coupling” and “tripartite interactions” from the revised manuscript, as they may cause misunderstanding. In the new version, our discussion is strictly based on experimentally and theoretically supported mechanisms, focusing on a physically grounded and verifiable picture: the strengthened antiferromagnetic order suppresses spin-allowed interlayer charge transfer, thereby weakening interlayer electron-hole recombination and its contribution to the optical transitions, which ultimately leads to a blueshift of the exciton emission.

Other possible factors that may influence PL – such as strain, thermal expansion, or magnetoelastic effects – have been systematically examined and excluded in Q1, and thus are not the primary origin of the observed PL energy shift.

Second, regarding the reviewer's comment that “*the PL of CrSBr exhibits significant vibrational fine structure (see Lin et al., ACS Nano 2024, 18, 2898)*” we appreciate this helpful suggestion and have included the citation in the revised manuscript. As reported by Lin et al., the vibrational fine structure appears prominently only at **very low temperatures (~4 K)**. When the temperature increases to **above ~80 K**, these features rapidly diminish or fully vanish, and the PL spectrum evolves into a single broad peak, consistent with earlier reports [*Nat. Mater.* 20, 1657-1662 (2021)]. Our study focuses on the spectral response of CrSBr **around its magnetic transition temperature (T_N)**, far above the temperature range where vibrational fine structure is resolvable. Therefore, vibrational fine structure is not expected to play a significant role in our experimental conditions and cannot account for the abrupt blueshift observed near T_N . Notably,

although line broadening at elevated temperatures increases the uncertainty of precise peak determination, the **overall spectral shift of the PL peak across a broad temperature range remains pronounced**. Additionally, our DFT calculations reveal a significant bandgap change under interfacial coupling, consistent with the experimental trend. This agreement further supports that the observed PL shift has a clear electronic-structure origin, rather than arising from fitting uncertainty or linewidth broadening.

Third, following the reviewer's suggestion, we have re-fitted the temperature dependence of the PL energy near the magnetic transition. The extracted slopes above and below T_N are -3.76×10^{-4} and -8.25×10^{-5} (Fig. R7), respectively, showing a clear difference and indicating that PL energy is sensitive to the establishment of magnetic order. Regarding the reviewer's comment that "for $T > T_N$ there are no coherent magnons to couple with," we fully agree. However, it is important to note that the AFM transition in CrSBr is not an abrupt jump but rather evolves gradually over a finite temperature window [*Nat. Mater.* 21, 754 (2022); *Adv. Mater.* 32, 2003240 (2020); *Nano Lett.* 21, 3511 (2021)].

In the CrSBr/FGT heterostructure, the strong intrinsic ferromagnetism of FGT can induce or stabilize short-range magnetic correlations in CrSBr via interfacial exchange coupling (as confirmed in Fig. R3), thereby extending the temperature range where magnetic correlations influence excitonic states. As mentioned in our previous response to Reviewer 3, this effect can reasonably explain the slight deviations near and slightly above T_N .

Consequently, the overall trend of the data in Fig. S5 agrees well with the fitted curve within the expected uncertainty. Small deviations near T_N may be attributed to short-range magnetic correlations and remain consistent with the unified spin-exciton interaction framework, without implying a new mechanism.

Regarding the presentation of the data, the black lines in Figs. 2a-b serve as manually drawn guide lines to visualize the overall evolution of the PL peak with temperature, and thus may exhibit slight offsets from the quantitatively extracted peak positions shown in Fig. 2d. Similarly, the arrows in Fig. 2c are schematic and not used for quantitative peak determination. In the revised manuscript, we have explicitly clarified this in the figure captions and redrawn the schematic lines and arrows with improved accuracy (Fig. 2). We emphasize that **the peak positions in Fig. 2d are the sole source used for quantitative analysis**.

Finally, we thank the reviewer for pointing out ambiguities in the fitting expressions in

Fig. S5 and Fig. S12. We have corrected the notation in the revised version. These fitting functions are phenomenological forms with broad applicability and can capture the characteristic asymptotic behavior near the magnetic transitions, they are primarily intended to describe the experimental trends associated with the magnetic transition.

Fig. R7. Variations of the PL peak positions for CrSBr/FGT under different temperatures. The two dashed lines around 130 K represent the slopes of the temperature dependence of the PL energy above and below T_N , respectively.

Revision:

Fig. 2. Photon properties of pristine CrSBr and CrSBr/FGT. **a-b**, Contour color map of the normalized PL intensity with varying temperatures for pristine CrSBr **(a)** and CrSBr/FGT **(b)**. The black lines are manually drawn guide lines to highlight the overall temperature-dependent evolution of the PL peak position. **c**, Temperature-dependent PL spectra for pristine CrSBr and CrSBr/FGT. The triangles correspond to the energy position of their PL peaks. The arrows indicate the general direction of the spectral shift. **d**, Variations of the PL peak positions for pristine CrSBr and CrSBr/FGT under different temperatures. **e**, The difference of PL peak shift between CrSBr and CrSBr/FGT as a function of temperature.

Fig. S5. Magnetic-order-dependent exciton fitting. Variations of the PL peak positions for CrSBr/FGT under different temperatures. The red solid curve is a best fit to the power law function $P^i(PL\ intensity) \propto [a + b(T - T_N)^\beta]^2$.

4. I do not see the emphasized finding of bidirectional modulation of PL peak energy to be very remarkable.

Cr(III) compounds show a range of PL energies. It can be expected that the PL peak energy can be modified by chemistry, strain, charge doping, etc. The stated novelty of some recent works in this area has been the observation that external fields may additionally be used to control the PL peak energy in a single sample at a fixed temperature. The present finding that the PL peak can also be tuned by interfacial effects is not in the same category as these recent works, because it cannot be controlled by external stimuli. The shifts are also a small percentage of the bandwidth, which limits the utility, as previous referees have noted.

Response: We are grateful for your thoughtful comments. CrSBr, as a two-dimensional van der Waals material that simultaneously exhibits magnetic and semiconducting properties, differs fundamentally from conventional Cr(III) compounds, and therefore the two cannot be directly compared. Traditional Cr(III) compounds, such as the Cr(III) compounds in rubies that you noted here, typically do not possess a stackable van der Waals structure. Therefore, they can neither do they provide the ability to form high-quality heterostructures with other 2D materials nor enable controllable interfacial spin-exciton interactions. In contrast, CrSBr stands out due to its quasi-1D magnetic ordering, strong magneto-optical correlations, and excellent compatibility with a wide variety of vdW materials. These attributes grant CrSBr clear advantages in excitonic manipulation, magneto-optical coupling, and spintronic applications.

It is precisely these unique material characteristics that make the modulation of CrSBr excitons scientifically meaningful and technologically valuable – far beyond what can be achieved in traditional Cr(III) – based systems (e.g. Cr(III) compounds in rubies). Correspondingly, research on CrSBr has grown rapidly in recent years, with notable progress reported across leading journals [*Nat. Nanotechnol.* 20, 617-622 (2025); *Nat. Mater.* 24, 1027-1033 (2025); *Nat. Photonics* 19, 1006-1012 (2025); *Phys. Rev. Lett.* 135, 136702 (2025)], demonstrating its strong potential in both fundamental studies and device development.

We acknowledge that the excitons in CrSBr can indeed be tuned using external magnetic fields, and this has been extensively documented. In contrast, our work provides a **more convenient and efficient modulation strategy** that operates **without the need for any external magnetic-field apparatus**, and the proposed heterostructure architecture is even more promisingly considered as a fundamental building block for

next-generation excitonic devices.

More notably, previous magnetic-field-based studies on CrSBr have only achieved **unidirectional redshifts** of the exciton energy. They could not realize the **bidirectional modulation** (both redshift and blueshift) demonstrated in our work. The dual-direction tunability presented here offers significant flexibility for optoelectronic device design and provides a useful reference for future wavelength-control strategies in laser and quantum photonic systems.

In summary, this heterostructure-based approach enables **simple, effective, and selective bidirectional modulation** of the CrSBr exciton PL energy, while also offering prospects for compact spintronic and opto-spintronic device architectures.

Regarding the reviewer's concern about "a small shift percentage of the bandwidth," we emphasize that **relative to an excitonic linewidth of ~100 meV**, the achieved **bidirectional modulation** of exciton energy is an even more crucial finding. Moreover, our bidirectional modulation amplitudes of **8.6% (redshift)** and **6.1% (blueshift)** are also considerable for exciton-energy-tuning research. We have responded to Reviewer #3 on this point previously and received complete satisfaction, as exemplified by: "*I thank the authors for putting the shift relative to total bandwidth in the abstract. I now am satisfied that my previous concerns have been met.*"

Here, we would like to emphasize once again that the key value and innovation of our work lie in two aspects:

- 1. The modulation scheme is expanded from conventional unidirectional tuning to bidirectional control**, enabling both redshift and blueshift of exciton energy – thus offering a flexible, on-demand tuning pathway.
- 2. The exciton-energy modulation is achieved purely through intrinsic interfacial interactions**, without requiring any externally applied magnetic or electric fields. This represents a distinct and desirable route toward compact, self-sustained opto-spintronic devices.

We believe that this **bidirectional tunability** is of greater scientific significance than the magnitude alone, as it provides critical degrees of freedom for future device engineering and potential wavelength control in quantum photonics and optoelectronics.

Regarding the comment "*I do not see the emphasized finding of bidirectional modulation of PL peak energy to be very remarkable,*" we would like to offer an objective clarification. Our manuscript has undergone rigorous evaluation by the Nature Communications editorial team and three full rounds of peer review. All reviewers have acknowledged the novelty and quality of our work. For example:

- *“I find the results rather interesting and all experimental data appears to be of high quality.”* (Reviewer #1)
- *“These findings hold significant importance for exciton physics and spintronics.”*
“I believe this work is interesting and suitable for publication in Nature Communications.” (Reviewer #2)
- *“Their experiments are of high quality and contain careful analysis.”*
“I find the experimental results valid and worth publishing.” (Reviewer #3)

In addition, we have substantially strengthened the manuscript by addressing previous reviewer concerns with rigorous analysis, and the reviewers ultimately expressed satisfaction with the revisions. We therefore remain confident that the results and conclusions in our work are innovative and will provide meaningful insights for future research and applications in related areas.

Finally, we sincerely thank the reviewer for raising these insightful comments. We have made every effort to address all concerns thoroughly and rigorously in this response letter. We hope that our explanations have clarified the issues you raised and will lead to a positive assessment of our work. We truly appreciate the reviewer’s careful evaluation and valuable contributions.

Response to Reviewers

Many thanks to the reviewers for their recognition of our manuscript (NCOMMS-25-47700-C) submitted to *Nature Communications*.

Title: Spin-exciton coupling modified by interfacial magnetic interactions in a van der Waals heterostructure
Yafei Chu, Lu Cheng, Chao Wang, Huijuan Wang, Minghui Fan, Zixun Zhang, Yuran Niu, Jheng-Cyuan Lin, Francesco Maccherozzi, Hengli Duan, Wensheng Yan

We sincerely thank Reviewer #4 for the positive evaluation and the recommendation to accept our manuscript (NCOMMS-25-47700-C). The reviewer's insightful suggestions and constructive comments have enabled us to present our results more rigorously and clearly, and we believe the manuscript has been significantly improved as a result.

We especially appreciate the reviewer for the time and effort they have devoted throughout the multiple rounds of review, as well as for acknowledging our efforts to address the concerns of all reviewers through comprehensive analyses and clarified descriptions.